# Leukocytospermia induces intraepithelial recruitment of dendritic cells and increases SIV replication in colorectal tissue explants

Mariangela Cavarelli [1✉], Stéphane Hua[1], Naima Hantour[1], Sabine Tricot[1], Nicolas Tchitchek[1,4], Céline Gommet[1,5], Hakim Hocini[2,3], Catherine Chapon[1], Nathalie Dereuddre-Bosquet [1] & Roger Le Grand [1]

Mucosal exposure to infected semen accounts for the majority of HIV-1 transmission events, with rectal intercourse being the route with the highest estimated risk of transmission. Yet, the impact of semen inflammation on colorectal HIV-1 transmission has never been addressed. Here we use cynomolgus macaques colorectal tissue explants to explore the effect of leukocytospermia, indicative of male genital tract inflammation, on SIVmac251 infection. We show that leukocytospermic seminal plasma (LSP) has significantly higher concentration of a number of pro-inflammatory molecules compared to normal seminal plasma (NSP). In virus-exposed explants, LSP enhance SIV infection more efficiently than NSP, being the increased viral replication linked to the level of inflammatory and immuno-modulatory cytokines. Moreover, LSP induce leukocyte accumulation on the apical side of the colorectal *lamina propria* and the recruitment of a higher number of intraepithelial dendritic cells than with NSP. These results suggest that the outcome of mucosal HIV-1 infection is influenced by the inflammatory state of the semen donor, and provide further insights into mucosal SIV/HIV-1 pathogenesis.

[1] Université Paris-Saclay, Inserm, CEA, Center for Immunology of Viral, Auto-immune, Hematological and Bacterial diseases (IMVA-HB/IDMIT), Fontenay-aux-Roses & Le Kremlin-Bicêtre, France. [2] Vaccine Research Institute - VRI, Hôpital Henri Mondor, Créteil, France. [3] Équipe 16 Physiopathologie et Immunothérapies dans l'Infection VIH, Institut Mondor de Recherche Biomédicale - INSERM U955, Créteil, France. [4] Present address: UMRS 959 Immunology-Immunopathology-Immunotherapy (i3), Sorbonne University and Inserm, Paris, France. [5] Present address: Sanofi R&D, Translational In vivo Models/In Vivo Research Center France/Veterinary Services, Centre de Recherche de Vitry/Alfortville, Vitry-sur-Seine, France. ✉email: mariangela.cavarelli@cea.fr

HIV-1 transmission in men who have sex with men most commonly occurs via colorectal exposure to infected semen. Receptive anal intercourse is also an underestimated contributor to heterosexual infections[1,2]. Tissue morphology, integrity, and the inflammatory state, as well as the distribution of several cell types within the mucosa, greatly influence viral transmission. CD4[+] T cells in the gastrointestinal tract are predominantly activated and well-differentiated to express a clear memory phenotype and constitute the main target for HIV-1 and SIV replication[3,4]. Moreover, an extensive network of resident innate immune cells with antigen-presenting function, such as dendritic cells (DCs) and macrophages, are potential targets for the incoming virus[5]. Colonic and rectal CCR5[+], DC-SIGN[+] DCs are subepithelial and found throughout the thickness of the mucosa[6,7]. We previously showed in vitro and ex vivo that human colonic CD11c[+], DC-SIGN[+], CCR5[+] DCs can extend dendrites containing HIV-1 to epithelial cells, as well as retract them, and thus transfer infection to CD4[+] T cells: a clear proof of principle of an HIV-1-DC uptake mechanism in the gut[8]. During anal intercourse, semen is delivered up to 60 cm up the colorectal tract[9] and thus extends the possible contribution of DCs to HIV-1 transmission.

It is well established that semen is more than merely a vehicle of HIV-1 particles, as it is a complex mixture of cells and biological factors, including cytokines and chemokines, that may affect HIV-1/SIV transmission and further influence host immune responses and susceptibility to infection[10–16]. The ability of seminal plasma, the acellular semen fraction, to enhance HIV-1 transmission has been proven by its capacity to recruit HIV-1 target cells, including macrophages, DCs, and T cells, to the female reproductive tract (FRT)[12,17–21] and to increase infectivity, even at low viral titers[22]. Moreover, seminal plasma levels of inflammatory cytokines have been shown to affect the activation state of the recipients' cells in the FRT mucosa, thus enhancing transmission[23]. However, the structure and environment of the gastrointestinal tract are different from that of the FRT, and the effect of semen deposition onto the colorectal mucosa has been less investigated. Moreover, none of the previous studies analyzing HIV-1 colorectal transmission took into consideration the level of inflammatory cytokines present in seminal plasma and did not assess the presence of an underlying immune response induced by seminal plasma treatment[24,25].

Inflammatory cytokines, including IL-8, IL-6, IL-1ß, are enriched in seminal fluids from leukocytospermic individuals[26–28] and we reported an increase in inflammatory molecules in semen associated with leukocytospermia and SIV infection in nonhuman primates (NHPs)[29,30]. Here, we investigated the effect of increased inflammatory molecule concentrations in seminal plasma due to leukocytospermia on SIV infection of macaque sigmoid colonic tissue ex vivo. We found that seminal plasma facilitates SIV transmission and replication by recruiting leukocytes to the subepithelial level of the lamina propria and attracting CD11c[+] DCs inside the intact intestinal epithelium. There was significantly greater enhancement of SIV infection and DC recruitment in the presence of elevated cytokine concentrations when explants were exposed to leukocytospermic seminal plasma (LSP) compared to normal seminal plasma (NSP).

## Results
### Upregulation of inflammatory cytokines in leukocytospermic seminal plasma of uninfected cynomolgus macaques. We assessed leukocytospermia in the semen of SIV-negative cynomolgus macaques by immunocytochemistry (Fig. 1a, b) and flow cytometry (Fig. 1c–i). The leukocytospermic threshold was set to 10,000 CD45[+] events acquired (Fig. 1d). A higher total leukocyte

content in leukocytospermic than normal animals was reflected by a significantly higher count of CD3[+] T cells ($p = 0.0025$), including both CD4[+] ($p = 0.0051$) and CD8[+] ($p = 0.0025$) cell subsets, macrophages ($p = 0.0025$), and granulocytes ($p = 0.0013$) (Wilcoxon rank-sum test, Fig. 1e–i).

There was a significantly higher concentration of a number of pro-inflammatory (IL-6, IL-8, IL-12/23, IL-13, IL-17a, IL-18, G-CSF, MIP-1ß, MCP-1, sCD40L, TGF-α, TNF-α, TGF-ß1, and VEGF) and immunoregulatory (IL-2, IL-10, IL-15) molecules in LSP than NSP samples (Wilcoxon rank-sum test, Supplementary Fig. 1). Conversely, there were no significant differences in the levels of IL-1ß, IL1RA, IL-4, IL-5, IFN-γ, MIP-1α, TGF-ß2, or TGF-ß3 between NSP and LSP. Principal component analysis (PCA) clearly distinguished the NSP from LSP samples (Fig. 2a). Furthermore, specific signatures were unraveled by hierarchical clustering analysis of the relative cytokine levels in the LSP and NSP samples (Fig. 2b). Due to the low amount of semen collected from each animal, a pool of NSP ($n = 4$) and LSP ($n = 7$) was made for further studies. Pooled seminal plasma samples were representative of high vs low inflammatory cytokine profiles (Fig. 2b and Supplementary Fig. 1).

**Seminal plasma is not toxic for the sigmoid tissue.** The potential cytotoxic effect of LSP on colorectal tissue was excluded by histological examination, with the explants exposed to LSP showing a similar morphological structure as control tissues (Fig. 3a). Treated explants presented an intact epithelium (Fig. 3b, c and Supplementary Fig. 2a) and mononuclear cells distributed evenly throughout the lamina propria. The evaluation of the permeability to FITC-dextran (FD4) in the presence or absence of LSP further substantiated the tightness between the cells after 2 or 4 h of treatment (mean $2.98 \pm 0.21$, $3.20 \pm 0.18$, and $3.15 \pm 0.23\%$ in medium, 25% LSP, and 50% LSP, respectively) (Fig. 3d). Exposure to LSP also did not significantly affect tissue viability ($0.86 \pm 0.06\%$ and $0.93 \pm 0.02\%$ after 2 h and $0.936 \pm 0.09\%$ and $0.99 \pm 0.16\%$ after 4 h of treatment with 25% or 50% LSP versus treatment with medium, respectively) (Fig. 3e). Similar results were obtained using an in vitro polarized colonic Caco-2 epithelial cells monolayer (Supplementary Fig. 2b–d).

**Enhancement of SIVmac251 replication in colorectal tissue explants is influenced by the cytokine profile of seminal plasma.** We first optimized the explant culture conditions (Supplementary Fig. 3) and the amount of virus needed to infect 100% of the explants while avoiding excessive viral replication and cytotoxicity (Supplementary Results and Supplementary Fig. 4a, b).

The tissue explants were incubated with SIVmac251, with or without 25% NSP or LSP, to test the hypothesis that a different inflammatory state of the seminal plasma may affect the susceptibility of the colorectal mucosa to SIV infection. SIVmac251 replication was significantly higher in explants ($n = 3$) treated with LSP from a single macaque (Supplementary Fig. 4c) and the result was confirmed using seminal plasma pools on tissues obtained from two additional donors (Fig. 4). Indeed, we observed significant differences in SIVmac251 replication at 9 and 12 days post-infection (dpi) for tissues incubated with SIVmac251 and LSP ($p = 0.0197$ and $0.011$, respectively, Friedman test) versus that of controls without LSP, whereas there was only a trend in the presence of SIVmac251 with NSP at 12 dpi ($p = 0.0583$ Friedman test, Fig. 4a). An analysis of the area under the curve (AUC) confirmed these differences between conditions ($p = 0.0003$, $0.0585$, and $0.0419$ in SIV + LSP vs SIV, SIV + NSP vs SIV, and SIV + LSP vs SIV + NSP, respectively, Friedman test). Treatment with LSP or NSP resulted in 3.8- or 1.6-fold (median) greater cumulative viral production at 12 dpi, respectively, than in

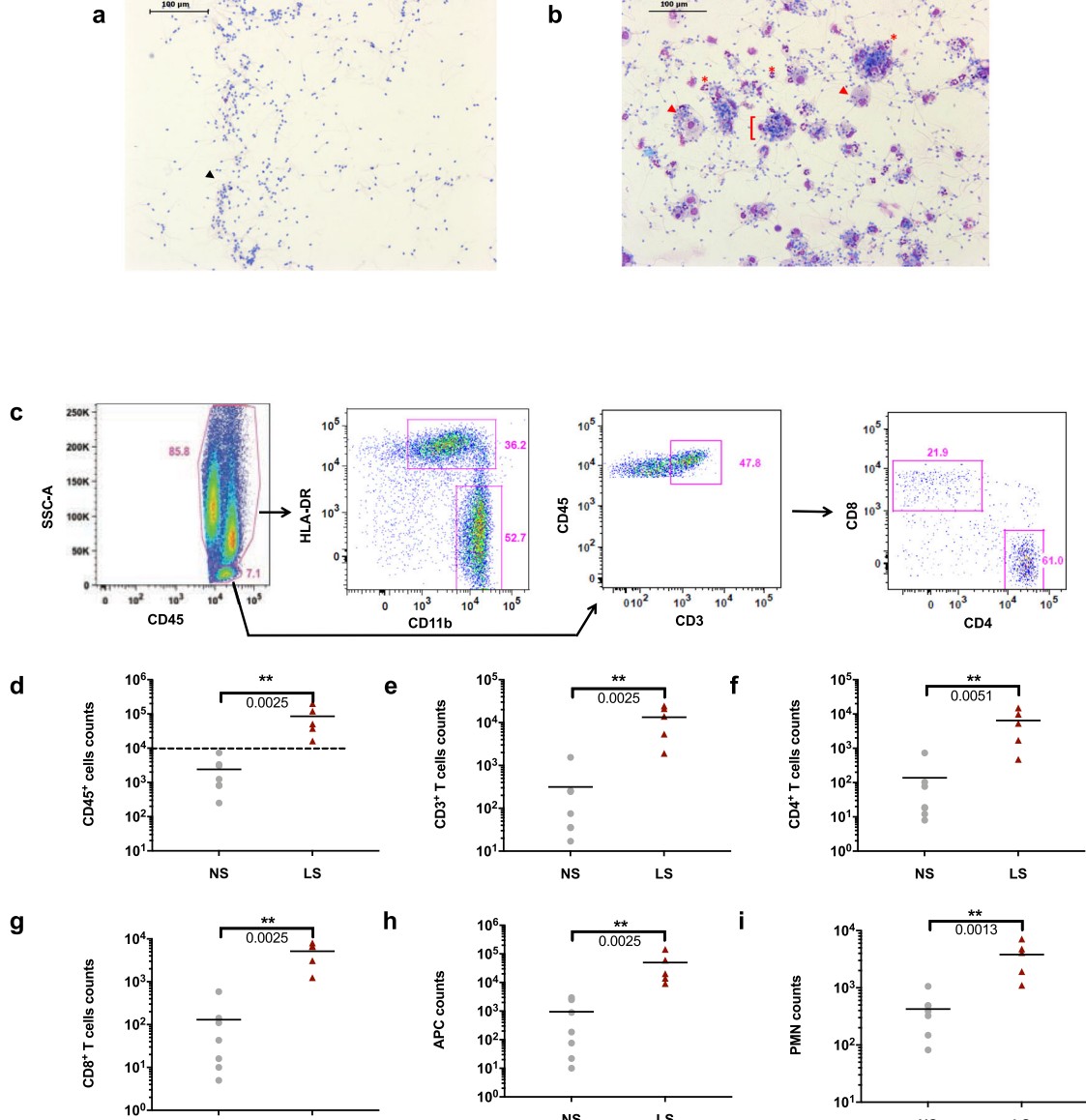

**Fig. 1 Characterization of normal and leukocytospermic semen of uninfected macaques. a**, **b** Representative immunocytochemical staining of cells from **a** normal semen (NS) and **b** leukocytospermic semen (LS). Numerous spermatozoa (black arrowhead) were present in NS. LS was rich in white blood cells, including lymphocytes (black *), macrophages (red arrowhead) and neutrophils (red *), frequently forming aggregates ([) with spermatozoa. **c** Gating strategy for semen leukocyte characterization. After the exclusion of doublets, cell debris, and dead cells, leukocytes are identified with the pan-leukocyte marker CD45. The SSC-A versus CD45 gate distinguishes lymphocytes from macrophages and polymorphonuclear cells on the basis of morphology. CD11b and HLA-DR distinguish HLA-DR$^{bright}$ CD11b$^{mid-to-bright}$ antigen-presenting cells (APC) from CD11b$^{bright}$ HLA-DR$^{neg-to-low}$ polymorphonuclear (PMN) cells. CD3$^+$ T cells are gated against CD45 and CD4$^+$ T cells are separated from CD8$^+$ T cells. **d–i** Number of events acquired by flow cytometry per sample, as all collected semen cells were analyzed for NS ($n = 7$) and LS ($n = 5$): **d** CD45$^+$ cells, **e** CD3$^+$ lymphocytes, **f** CD4$^+$ T cells, **g** CD8$^+$ T cells, **h** APC, and **i** PMN. The dotted line in panels (**d**) represents the leukocytospermia threshold (10,000 CD45$^+$ events acquired). The mean is shown. Statistical significance between groups was tested using Wilcoxon rank-sum tests, **$p < 0.01$.

SIVmac251 infected non-treated explants ($p = 0.0022$ and 0.0476, respectively, Wilcoxon rank-sum test, Fig. 4b). In addition, explants treated with seminal plasma showed an accumulation of SIV DNA (median number of SIV DNA copies = 257,500 with LSP, 199,000 with NSP, and 90,229 for SIVmac251 control cultures) (Fig. 4c), confirming that higher virus production is associated with a higher number of infected cells. To dissect whether the increased SIV replication was linked to higher level of seminal plasma inflammatory and immunoregulatory cytokines we analyzed the association between cytokine concentration in seminal plasma and explants infection. Viral replication was

positively correlated with a number of cytokines and chemokines. The strongest correlation ($p < 0.001$ and $r > 0.6$) was observed with IL-2, IL-5, IL-10, IL12/23, sCD40L, and VEGF (Spearman correlation, Table 1). Overall, these results show that seminal plasma favors ex vivo SIV infection of the colonic mucosa and that the enhancement of infection is influenced by the inflammatory factors present in seminal fluids.

**Exposure to seminal plasma activates lamina propria lymphocytes.** In order to identify the SIV target cells in the colonic explants and evaluate the effect that seminal plasma exposure

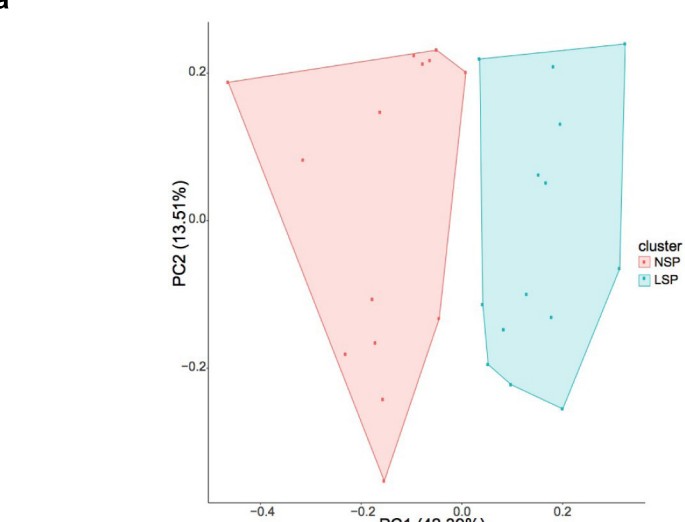

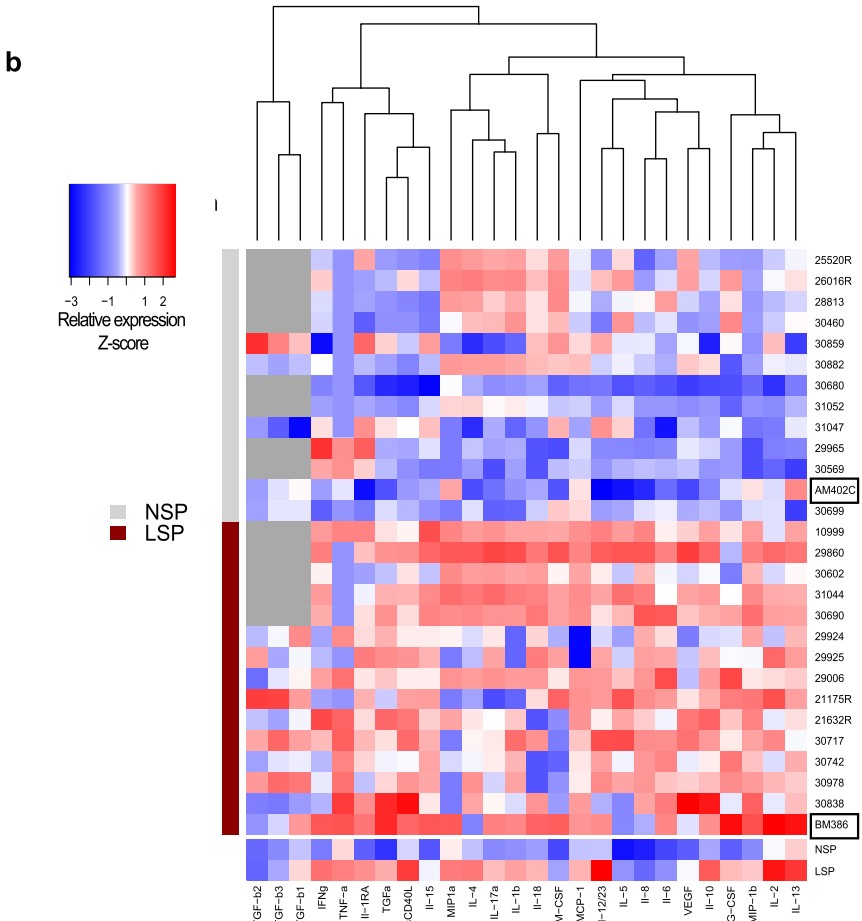

**Fig. 2 Influence of leukocytospermia on cytokine levels in the seminal plasma of uninfected macaques. a** Principal component analysis (PCA) showing that normal (NSP, pink dots) and leukocytospermic seminal plasma (LSP, blue dots) samples cluster separately. **b** Heatmap from Luminex data of 26 cytokines and chemokines in seminal plasma from 13 normal (NSP) and 15 leukocytospermic (LSP) uninfected macaques (with the exception of TGF-ß 1-2-3, for which 5 NSP and 10 LSP were evaluated) and comparison with the seminal plasma pools (NSP and LSP) shown at the bottom of the heatmap. Boxed codes represent individual NSP and LSP used for ex vivo infection experiments. Data are shown as relative molecule concentration compared to the mean value. Upregulated molecules are shown in red and downregulated molecules in blue.

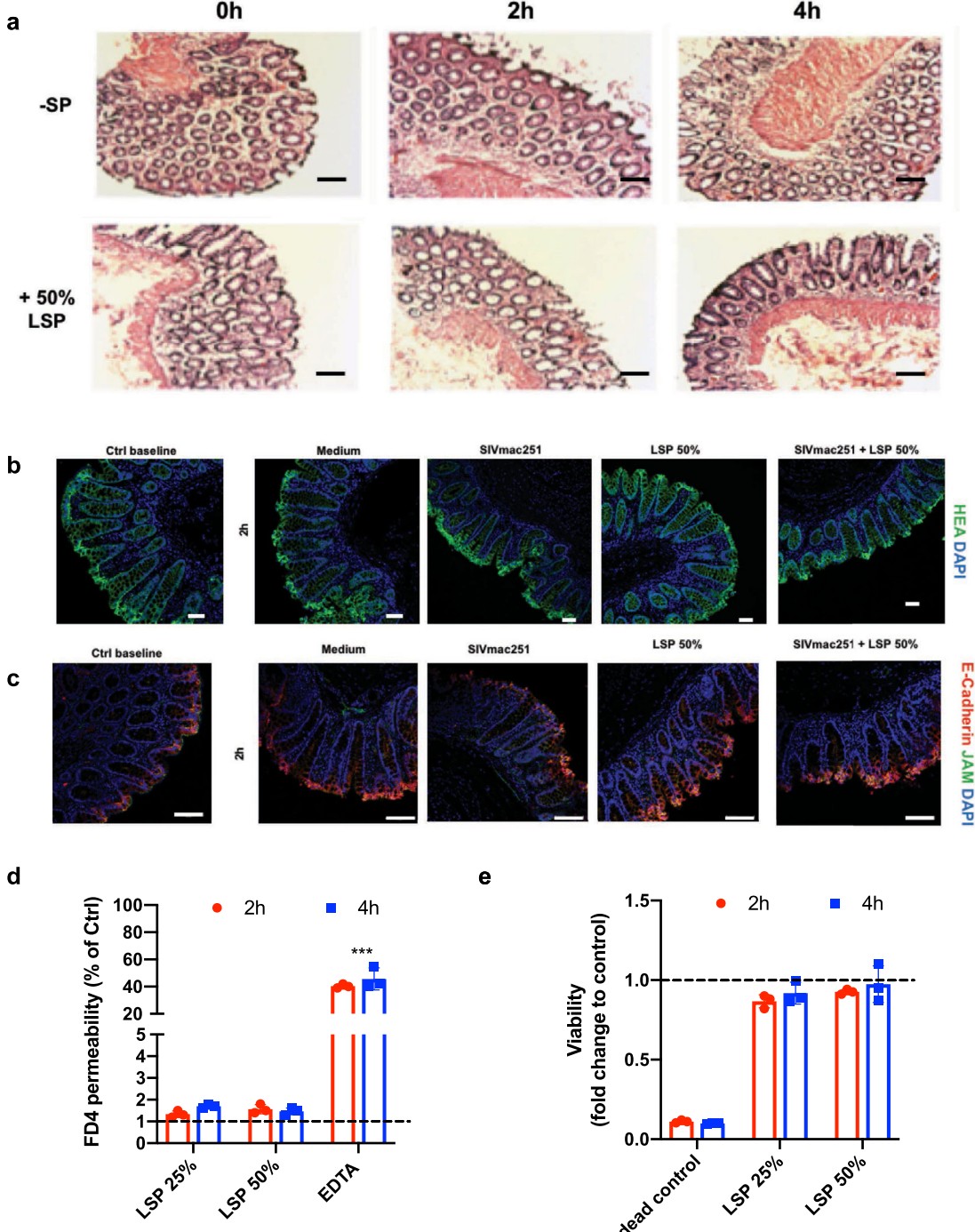

**Fig. 3 Ex vivo evaluation of seminal plasma toxicity. a** Representative histology (H&E staining) of the explant before and after 2 and 4 h of culture with or without 50% LSP, showing intact epithelium. Magnification ×20. Scale bar = 50 μm. **b**, **c** Immunofluorescence staining for the human epithelial antigen (HEA-FITC, green), E-cadherin (red), and the junctional adhesion protein (JAM, green) of explants before and after treatment for 2 h with culture medium (negative control), SIVmac251 with or without 50% LSP, and 50% LSP. Scale bar = 50 μm in (**b**) and 100 μm in (**c**). **d** Integrity of the epithelial barrier measured by the addition of Dextran-FITC (FD4, 4 kDa, 250 μg/ml) to the apical side of the explant with or without 25% or 50% LSP for 2 and 4 h. Results are shown as the percentage of positive control (dye added to the basal medium at the beginning of the experiment). EDTA (100 mM) was included as a control of barrier disruption. The statistical significance between conditions was tested using Wilcoxon signed-rank tests, ***$p < 0.0005$. **e** Viability of the explants exposed, or not, for 2 and 4 h to 25% or 50% LSP measured by the MTT assay. A solution of 10% bleach was included as a control for death. Results are shown as the mean ± SD of triplicates from a representative experiment of three.

might have on their phenotype, lamina propria mononuclear cells (LPMCs) were isolated from the sigmoid tissue and exposed during 2 h to either complete medium, LSP or NSP. Immuno-phenotyping was performed after 72 h of culture. The gating strategy is shown in Supplementary Fig. 5a. Most colonic CD4[+] T cells and CD8[+] T cells were central memory cells (CD28[+]/CD95[+], 49.4 ± 7.8% and 40.3 ± 2.9%, respectively) (Fig. 5a). A higher proportion of effector memory cells (CD28[−]/CD95[+]

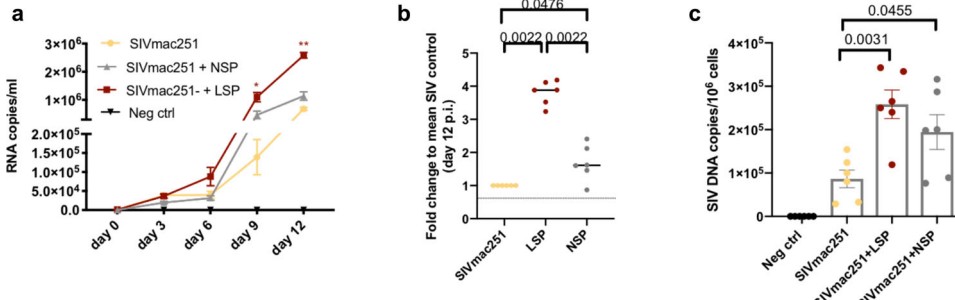

**Fig. 4 Increased SIVmac251 replication in sigmoid tissue explants is associated with seminal plasma cytokine content.** Sigmoid colon explants were infected with SIVmac251 with or without a pooled NSP or LSP. **a** Kinetics of SIVmac251 replication in explants treated with virus alone (yellow line), virus plus 25% LSP (red line), or virus plus 25% NSP (gray line). Culture medium was used as negative control. Virus replication was evaluated in the basal supernatant as viral RNA copies/ml. **b** Fold change increase in cumulative viral production by each ($n = 6$) biopsy measured by RT-PCR compared to the mean positive control value (explant exposed to SIVmac251 alone, mean of 6 values). **c** SIV DNA quantification in explants infected in the presence or absence of 25% LSP or 25% NSP, harvested at the end of the culture (day 12). Results shown in panels (**a–c**) are the mean ± SEM of triplicates from two independent experiments. Statistical significance between the different conditions was tested using Friedman tests with post hoc Benjamini, Krieger, and Yekutieli tests, except for panel (**c**), for which Wilcoxon rank-sum tests were used.

| Table 1 Correlations between the level of cytokines and chemokines in seminal plasma and viral replication at day 12 p.i. | | |
| --- | --- | --- |
| **Molecules** | **p-value** | **r-value** |
| IL-1ß | <0.0001 | 0.58 |
| IL-1RA | <0.0001 | 0.58 |
| IL-2 | <0.0001 | 0.63 |
| IL-4 | <0.0001 | 0.41 |
| IL-5 | <0.0001 | 0.72 |
| IL-6 | 0.0007 | 0.24 |
| IL-8 | 0.0003 | 0.26 |
| IL-10 | <0.0001 | 0.72 |
| IL-12/23 | <0.0001 | 0.61 |
| IL-13 | <0.0001 | 0.51 |
| IL-15 | 0.0002 | 0.28 |
| IL-17a | <0.0001 | 0.49 |
| IL-18 | 0.0003 | 0.26 |
| G-CSF | 0.0009 | 0.23 |
| GM-CSF | 0.0003 | 0.26 |
| IFN-γ | <0.0001 | 0.59 |
| MCP-1 | <0.0001 | 0.4 |
| MIP1-α | <0.0001 | 0.44 |
| MIP-1ß | 0.0012 | 0.22 |
| sCD40L | <0.0001 | 0.74 |
| TGF-α | <0.0001 | 0.31 |
| TNF-α | <0.0001 | 0.32 |
| TGF-ß1 | <0.0001 | 0.33 |
| TGF-ß2 | 0.0107 | 0.14 |
| TGF-ß3 | 0.0001 | 0.29 |
| VEGF | <0.0001 | 0.73 |

/CD45RA⁻) was found among CD4+ T cells (27.2 ± 4.1% vs 17.5 ± 3.8%, respectively), whereas naive cells (CD28⁺/CD95⁻) were mostly present among CD8⁺ T cells (15.3 ± 4.1% and 32.6 ± 5.4%, respectively). Effector memory T cells re-expressing CD45RA (TEMRA, CD28⁻/CD95⁺/CD45RA⁺) were rare among CD4⁺ T cells (1.2 ± 0.5% and 6.7 ± 1.5%, respectively) (Fig. 5a). Treatment with both LSP and NSP did not change the relative proportion of each subset among CD4⁺ and CD8⁺ T cells (Fig. 5a). Compared to medium-treated cells, CD4⁺ T cells and CD8⁺ T cells exposed to both LSP and NSP expressed higher level of HLA-DR ($p = 0.0033$ and $p = 0.0355$ for LSP and NSP in CD4 T cells, respectively, and $p = 0.005$ and $p = 0.0355$ for LSP and

NSP in CD8 T cells, respectively, Friedman tests, Fig. 5b, e), whereas a significant increase of CD69⁺ cells was observed only among CD4⁺ T cells ($p = 0.0153$ and $p = 0.0213$ for LSP and NSP, respectively, Friedman test, Fig. 5c, f). Interestingly, highly activated CD38⁺HLA-DR⁺CD69⁺CD4⁺ T cells were induced by both treatments ($p = 0.0015$ and $p = 0.0292$ for LSP and NSP in CD4 T cells, respectively, and $p = 0.0033$ and $p = 0.0077$ for LSP and NSP in CD8 T cells, respectively, Friedman tests, Fig. 5d, g). Those cells represent specific T-cell subset that is preferentially eliminated in primary SIV infection[31].

**Seminal plasma treatment induces leukocyte migration and recruits dendritic cells inside the colorectal epithelium.** An influx of leukocytes to the FRT following exposure to semen has been proposed as a mechanism mediating HIV-1 transmission, with macrophages and DCs being the most abundant recruited cells[12,17–21]. Since we were restricted by the size of the sigmoid colon, emphasis was placed on assessing total leukocytes recruitment to the colorectal mucosa following LSP treatment. Ex vivo dynamic imaging of intestinal leukocytes by time-lapse confocal microscopy (Supplementary Fig. 6a) showed massive infiltration of CD45⁺ cells in LSP-treated explants (Fig. 6a and Supplementary Fig. 6b), and a significantly higher cell number/mm² compared to medium (negative control) and SIVmac251 (positive control) treated explants ($p < 0.0001$ and $p = 0.0108$, respectively, Wilcoxon signed-rank test Fig. 6b). The infiltration of CD45⁺ cells occurred as soon as 1 h post-exposure to LSP and the difference versus the controls was maintained until the last time point analyzed (Fig. 6c and Supplementary Fig. 6b). These results show that LSP exerts a strong chemotactic effect on leukocytes, and is more efficient than the virus itself in inducing immune cells migration toward the apical side of the mucosa.

Given the observed chemotactic effect on leukocytes, we tested the hypothesis that intraepithelial recruitment of early HIV target cells, such as DCs, might be a mechanism favoring LSP versus NSP mediated virus transmission. Indeed, we previously reported that intestinal DCs migrate from the lamina propria inside the colonic epithelium following exposure to R5 HIV-1 through a CCR5-dependent mechanism to capture the virus and transfer the infection to receptive CD4⁺ T cells[8].

Ex vivo apical stimulation of macaque tissue explants with SIVmac251, 25% LSP, or 25% NSP induced lamina propria resident HLA-DR⁺ (Fig. 7a, c) CD11c⁺ (Fig. 7b, d) DCs to migrate between epithelial cells, although the latter one was less efficient.

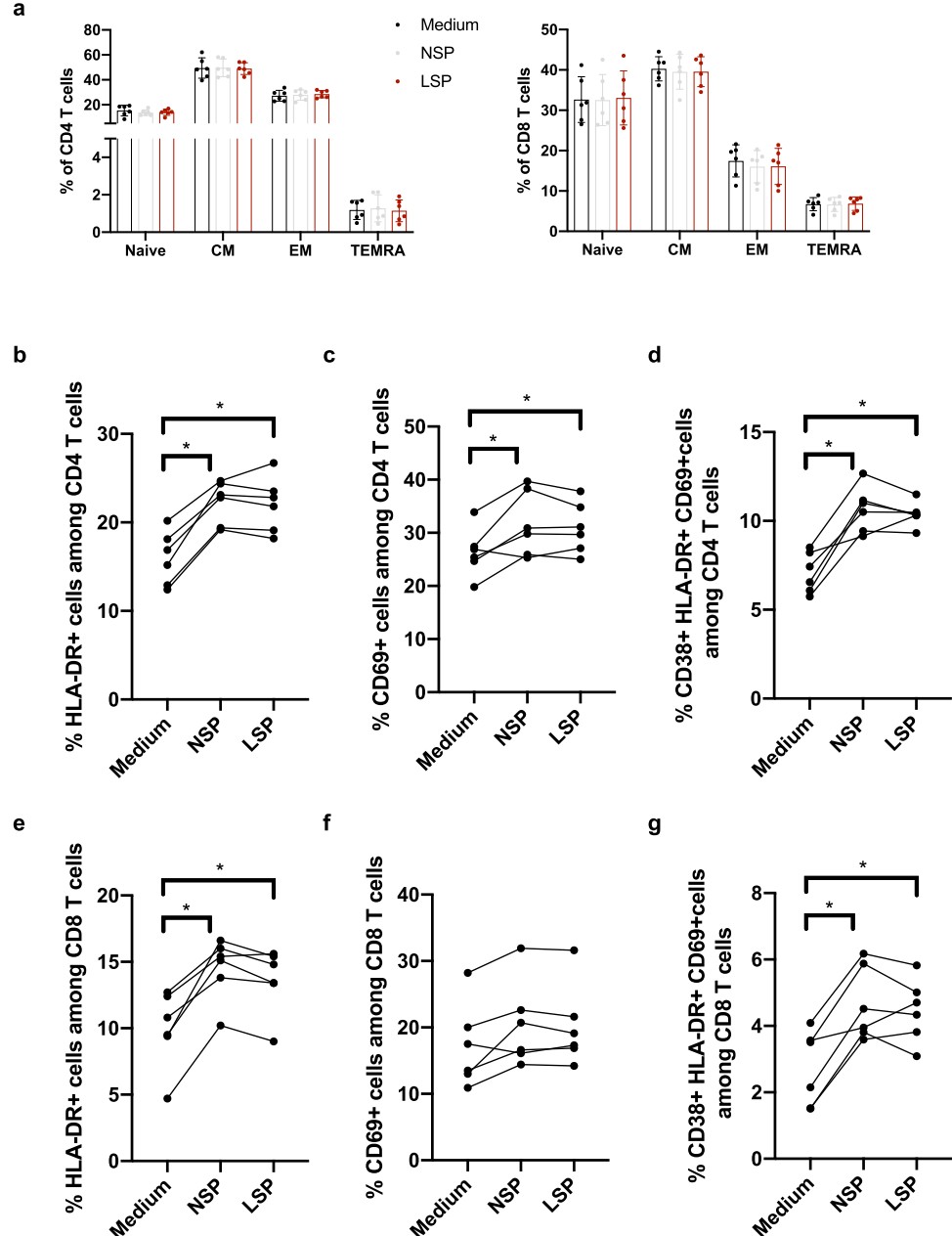

**Fig. 5 Phenotype and activation status of leukocyte populations present in the sigmoid colon before and after exposure to seminal plasma.** Lamina propria mononuclear cells isolated from six different donors were incubated 2 h with medium, 25% NSP or 25% LSP, washed, and immunophenotyped 72 h later. **a** The graph shows the proportion of naive, central memory (CM), effector memory (EM), and effector memory RA (TEMRA) cells among colonic CD3+CD4+ and CD3+CD8+ T cells. Each symbol represents one animal and the mean ± SD is shown. **b–g** Expression of the activation markers HLA-DR (**b**, **e**), CD69 (**c**, **f**) and highly activated CD38+HLA-DR+CD69+ cells (**d**, **g**) by CD3+CD4+ (**b–d**) and CD3+CD8+ T cells (**e–g**) from the colon. Each symbol represents one animal. $p < 0.05$ denotes significant difference between conditions (Friedman tests with post hoc Benjamini, Krieger, and Yekutieli tests).

Intraepithelial cells were not detected in medium-treated explants. Quantification of intraepithelial HLA-DR+ and CD11c+ cells revealed a further increase in DC migration relative to that in the presence of the virus alone when LSP was mixed with SIVmac251 ($p = 0.0044$ and $0.0012$ for HLA-DR+ and CD11c+ cells, respectively, Kruskal–Wallis test), whereas an additive effect was not observed in presence of NSP (Fig. 7c, d). These results were confirmed using an in vitro dual-chamber Caco-2/DC co-culture model (Supplementary Fig. 7).

We then asked whether the differential DC migration observed in LSP versus NSP-treated explants could be due to the higher concentration of CCR5-binding molecules found in the former one (Fig. 2b). LPMC were isolated from the sigmoid colon and exposed ex vivo during 2 h to both LSP and NSP and rested for another 4 h. DCs were identified among live CD45+ cells as lineage-HLA-DR+CD11c+CD64−CD123−CD14−CD16− cells (Supplementary Fig. 5b). While some variation was observed between donors ($n = 6$), compared to medium-treated cells, a tendency toward an upregulation of CCR5+ cells was observed following exposure to NSP (1.7 mean fold), whereas a down-regulation was induced by LSP treatment (0.6 mean fold). A statistical significative difference was observed when comparing

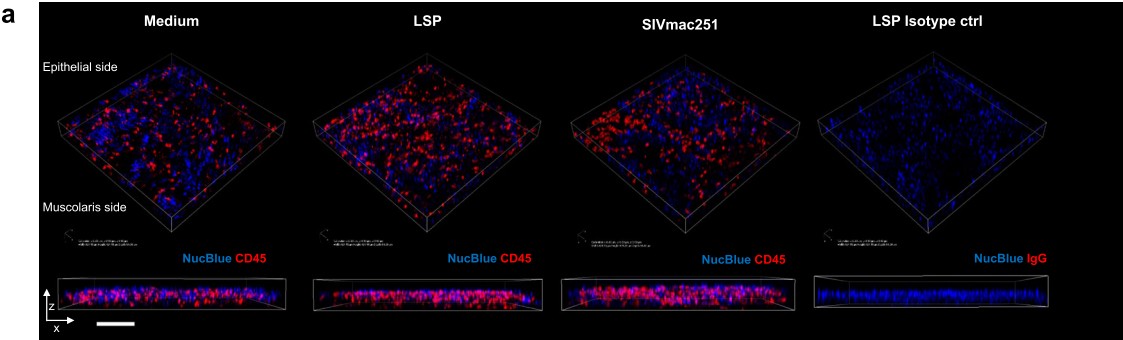

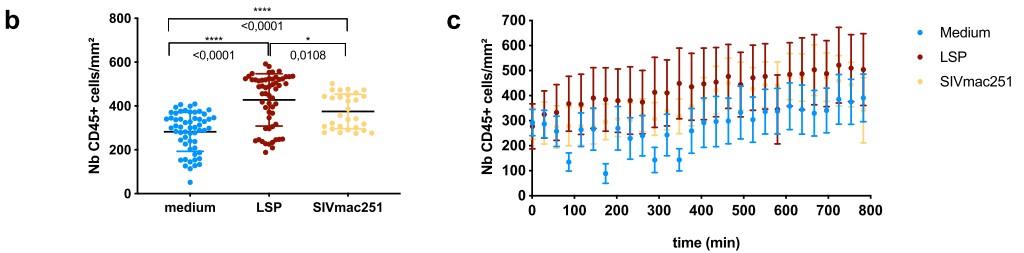

**Fig. 6 Imaging of leukocyte recruitment by seminal plasma by time-lapse microscopy. a** Three-dimensional rendering of representative fields obtained using NIS-Elements AR Analysis 5.02.0 (NIKON) and ImageJ software. Mouse anti-human CD45 antibody shows the leukocytes (red) and NucBlue dye label the nuclei (blue). Explants were either exposed to medium (without LSP), SIVmac251, or 25% LSP. An explant incubated with an isotype control antibody and exposed to LSP is shown as a negative control. A representative image obtained after 600 min of acquisition is shown. Scale bar = 25 µm. Experiments were repeated three times. **b** Number/mm² of CD45⁺ cells imaged during 14 h of acquisition in medium, LSP, and SIV-treated explants. Each dot represents the number of cells/mm² imaged in the z-stack. Mean and SEM of three experiments are shown. Statistical analysis was performed using Wilcoxon signed-rank tests. **c** Number/mm² of CD45⁺ cells imaged every 29 min in the presence (red symbols) or absence (blue symbols) of seminal plasma as well as SIVmac251 (yellow symbols). The mean and SEM of three experiments are shown.

LSP vs NSP exposed cells ($p = 0.0039$, Friedman test, Fig. 7e). Conversely, a modulation of CCR5 by CD4⁺ T cells exposed to NSP and LSP was not observed (Fig. 7f). These results suggest that increased concentration of CCR5-binding chemokines in LSP may account for increased CD11c⁺ DC recruitment inside the intact colonic epithelium, a mechanism that might facilitates SIV transmission.

## Discussion

We took advantage of the SIV model and explant culture conditions, which recapitulate the main features of in vivo exposure of the colorectal mucosa to semen, to investigate whether the levels of inflammatory cytokines present in seminal plasma influence the likelihood of SIV/HIV infection. Semen cytokine levels may show heterogeneity, irrespective of HIV status, possibly linked to genital inflammation[11,26]. Previous studies considered the cytokine/chemokine content of semen to be homogeneous and did not evaluate the influence of these seminal plasma factors on the local response of the colorectal mucosa[24,25]. Here, we explored the contribution of inflammatory and immune-modulatory cytokines present in normal and leukocytospermic semen to mucosal SIV infection.

As described in human semen, the cytokine profile of macaque LSP was distinct from that of NSP samples, as shown by the strong upregulation of several molecules (IL-1ß, IL-6, IL-8, IL-15, G-CSF, GM-CSF, and MCP-1)[32,33]. Among them, MCP-1 is a chemoattractant for monocytes, memory T cells, DCs, and NK cells[34] and has been shown to play a role in recruiting immune cells to the FRT following ejaculation[20]. IL-8, GM-CSF, and IL-15 are involved in the recruitment, maturation, and proliferation of monocytes, T and B cells, DCs, and NK cells at potential sites of inflammation[35–37]. Thus, inflammatory semen contains high concentrations of immune mediators that are involved in the recruitment, maturation, and survival of immune cells. Under conditions of infection, these mediators may be involved in local viral replication and the associated increased risk of viral transmission. Specifically, the pro-inflammatory cytokine G-CSF attracts and promotes the survival of neutrophils, influences T-cell function, and activates DCs[35,38]. IL-8 may promote HIV-1 infection[39]. IL-6 and IL-8 can directly upregulate HIV-1 gene transcription and increase the activation and life-span of HIV-infected or bystander target cells[40,41]. Moreover, transient induction of IL-8 and IL-1ß by cervical cells following exposure to semen or seminal plasma was suggested to have implications for HIV risk in women[12,42]. Future NHP studies or cohort studies of HIV-1 patients evaluating the effect of infected semen are also warranted, since LSP from uninfected macaques might not be fully representative of SIV-infected seminal plasma.

A strength of our ex vivo transmission model is that it closely mimics the biology of HIV/SIV sexual transmission and the natural route and dynamics of viral entry into the mucosa. The use of a non-polarized explant system probably explains why no effect of seminal plasma on HIV-1 colorectal transmission was observed in a previous study[24]. In addition, cytotoxicity, a laboratory artifact representing one of the major limitations of working with seminal plasma, was not observed under our experimental conditions. In macaques, SIV has been shown to infect rectal target cells within 1–4 h post-exposure[43,44]. Thus, we focused on the relevant time-frame of 2–4 h to study the impact of seminal plasma on viral transmission. In accordance with the results of previous ex vivo[24] and in vitro[45] studies, we did not observe any impact of seminal plasma on the integrity of the epithelial cell barrier in either of our models.

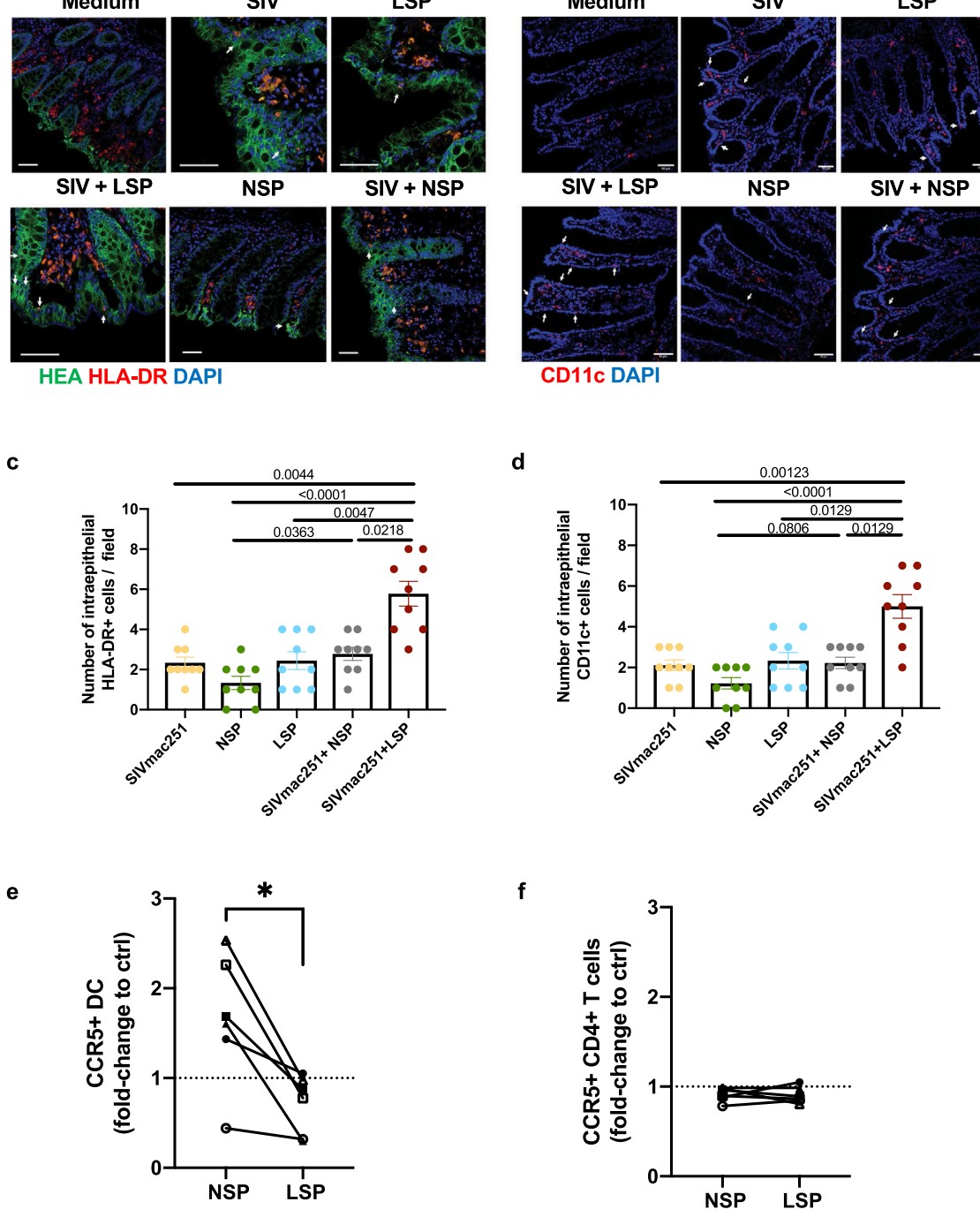

**Fig. 7 Seminal plasma favors dendritic cell migration inside the colonic epithelial barrier. a, b** Immunofluorescence staining for the human epithelial antigen (HEA-FITC, green), HLADR+ cells (red, **a**), and CD11c+ cells (red, **b**) after 1.5 h of treatment of the macaque colonic explant with medium, SIVmac251, 25% LSP, SIVmac251 + 25% LSP, 25% NSP, or SIVmac251 + 25% NSP. Nuclei are stained with DAPI. Arrows point to intraepithelial HLA-DR+ or CD11c+ cells. Scale bar = 50 µm. **c, d** Quantification of the number of HLA-DR+ cells (**c**) and CD11c+ cells (**d**) present inside the intestinal epithelium following exposure to the different stimuli. Bars represent the mean ± SD of three randomly chosen fields from three different experiments. Statistical significance between the different conditions was tested using Kruskal–Wallis tests with post hoc Benjamini, Krieger, and Yekutieli tests. **e, f** Lamina propria mononuclear cells isolated from 6 different donors were incubated 2 h with medium, 25% NSP or 25% LSP, washed, and immunophenotyped 6 h later. Percentage of CCR5+HLA-DR+CD11c+ DC (**e**) and CCR5+CD4+ T cells (**f**) exposed to NSP and LSP is expressed as fold change compared to the control (medium). $p < 0.05$ denotes significant difference between conditions (Friedman tests with post hoc Benjamini, Krieger, and Yekutieli tests).

By analogy with other studies using human cervico-vaginal and colorectal explant cultures[12,24,46,47], long-term culture led to alteration of the colonic tissue, irrespective of seminal plasma or viral treatment. Although degeneration could affect the quality of the living, infectable cells, the presence of infected cells within explants at the end of the culture was confirmed by SIV DNA

quantification, suggesting that the fraction of SIV target cells residing in the mucosa are sufficient to support viral replication and that modulation by seminal plasma occurred.

A major finding of our study is that LSP enhances SIVmac251 infection, whereas NSP is less efficient, as shown by differences in culture supernatant viral load. As described by others, we observed cells emigrating from tissues as soon as 24 h after culture[48]. We do not exclude the possibility that a higher number of SIV-infected cells might have exited explants upon treatment with LSP compared to NSP, thus explaining the non-significant difference in the number of infected cells at 12 days p.i. in the two treatment conditions. We accounted for intra-donor variation of seminal plasma cytokine content by performing independent experiments using either NSP and LSP from an individual macaque or a pool of NSP and LSP. In the absence of breaches in the mucosa, the early mechanisms involved in crossing of the intestinal epithelium by HIV/SIV imply the recruitment of immune cells that possess migratory properties, including lymphocytes, macrophages, and DCs. They can be considered as the first target cells. Due to their proximity to the luminal environment and constant exposure to a myriad of food and microbial antigens, mucosal CD4$^+$ T cells in the gastrointestinal tract have been described to be predominantly activated and expressing a memory phenotype[49]. In agreement with these observations, we found most CD4$^+$ T and CD8$^+$ T cells of the macaque sigmoid mucosa to be central memory cells. These cells are prone to infection with HIV and, once depleted, do not repopulate the lamina propria, as shown in both untreated and treated patients and SIV-infected macaques with progressive disease[3,4,31,50–52]. Direct exposure of LPMCs to both NSP and LSP, did not change the proportion of CD4 and CD8 T-cell subsets, and induced an activated memory phenotype of CD4$^+$ T cells. Of note, our immunophenotype analysis was restricted to 72 h post culture. Accordingly, a significative difference in viral replication was not observed at 3 dpi in explant exposed to NSP and LSP, which instead became evident at 9 dpi. It is conceivable that the complex mucosal environment of the explants responding to NSP vs LSP could differentially affect expression of CD4$^+$ T cells activation markers at this later time point, and further experiments are required to substantiate this hypothesis.

It is also well established that both sperm and seminal plasma elicit transient infiltration of the cervix mucosa by leukocytes[12,19,53], suggesting a role for seminal plasma in the recruitment of immune cells. We primarily evaluated the effect of LSP on the mucosal barrier and cell chemotactic properties, independently of virions. We confirmed and extended these published data, established for the FRT, by assessing the increased number of total leukocytes in the lamina propria of LSP-treated explants versus controls by confocal time-lapse microscopy, which allowed us to measure cell migration within a 3D physiological context. Specifically, our results show an accumulation of CD45$^+$ cells toward the apical side of the lamina propria. The accumulation of monocytes and neutrophils following FRT exposure to seminal plasma has been reported, in accordance with our findings[12]. Although leukocytes migration by NSP was not assessed in this experimental setting, differential chemotaxis of HLA-DR$^+$, CD11c$^+$ DC, one of the first cell types to contact HIV-1 and playing an active role in capturing the virus[8], was observed when analyzing explants treated with LSP vs NSP. DC migration across the tight intestinal epithelium is triggered by the R5 viral envelope, which engages DCs via the CCR5 molecule. Here, we show that DCs respond to SIV treatment in a similar way. Interestingly, the absence of migration in Caco-2/DC cultures exposed to the supernatant of CEMx174 cells confirmed that DC chemotaxis occurs in response to the virus itself and not the cytokines/chemokines present in the viral inoculum. In

addition, the higher number of migrating cells detected when LSP, but not NSP, was mixed with SIV, is supporting the enhancing role of LSP. Because a higher number of CCR5-binding molecules have been found in LSP than NSP, it is possible that these mediators are responsible for the differential DC migration also observed under these in vitro and ex vivo conditions. Accordingly, CCR5 downregulation was observed in LSP vs NSP-treated lamina propria DCs, as indirect evidence of cognate chemokine binding to the receptor. Ex vivo assessment of CCR5 downregulation in seminal plasma-treated explants will be an important consideration for future studies, as to substantiate our observation. Moreover, our findings support further studies to evaluate the use of CCR5 inhibitors for the prevention of rectal HIV-1/SIV transmission, a strategy known to be effective in preventing vaginal transmission of SHIV in Rhesus macaques[54,55].

In conclusion, we provide a comprehensive study demonstrating that the inflammatory and chemotactic profile of seminal plasma may influence SIV uptake and replication ex vivo in the intestinal mucosa. Further validation and in-depth mechanistic studies on these pathways through SIV/NHP in vivo models, should provide further basic insights on mucosal HIV-1 pathogenesis. Such knowledge could help to reveal novel therapeutic targets to counteract HIV-1 infection and inflammation in the gastrointestinal tract.

## Methods

**Ethical statement and animal care**. Cynomolgus macaques (*Macaca fascicularis*), weighing 5–11 kg, were imported from Mauritius and housed according to European guidelines for animal care (Official Journal of European Union, L 358, December 18, 1986 and new directive 63/2010). All work related to animals was conducted in compliance with institutional guidelines and protocols approved by the local ethics committee (Comité d'Ethique en Experimentation Animale de la Direction des Sciences du Vivant au CEA under numbers: 2015032511332650 (APAFIS#373); 10-062; 12-103). The sigmoid colon was collected at necropsy.

Healthy human peripheral blood mononuclear cells (PBMCs) were obtained from buffy coats provided by the Établissement Français du sang (EFS). The human donors signed an informed consent form. All procedures were performed in accordance with the ethical standards of the institutional and the regional ethical committee.

**Semen collection, seminal plasma, and cell preparation**. All cynomolgus macaques tested negative for antibody responses to SIV, simian retrovirus type D (SRV), and simian T-cell lymphotropic virus (STLV) at the beginning of the study. Semen was collected from sedated animals following 5 mg/kg intra-muscular injection of Zoletil®100 (Virbac, Carros, France). Electroejaculation was performed by intrarectal electrostimulation of the prostate using a probe (12.7 mm diameter) lubricated with a conductor gel and an AC-1 electro-ejaculator (Beltron Instrument, Longmont, USA)[29]. Sequential stimulations were performed, with a pattern of 6 cycles, each cycle consisting of nine 2-s stimulations followed by a tenth stimulation lasting 10 s. The voltage was increased every two cycles (1–3 V for the first two cycles, 2–4 V volts for the third and fourth cycles, and 6–8 V for the last two cycles). If a complete ejaculate had not been obtained after six cycles of stimulation, a 7th cycle of stimulation at 7–10 V was performed. Total ejaculate was centrifuged 15 min at 775 × $g$ immediately after collection to separate the acellular fraction (seminal plasma) from the cellular fraction (semen cells). The cells were maintained at room temperature for a maximum of 1 h, centrifuged 10 min at 1500 × $g$, filtered through a 70-μM sieve, and washed with 5 ml PBS supplemented with 10% FCS. Seminal plasma was filtered, aliquoted and stored at −80° until use. For the constitution of the pools, 500 μl of seminal plasma from seven leukocytospermic and four normal animals were mixed. pH of the seminal plasma pools ranged between 7.2 and 7.4.

**Phenotypic characterization of semen leukocytes and identification of leukocytospermic animals**. Seminal cell staining was performed after saturation of Fc receptors by healthy macaque serum (in-house production) for 1 h at 4 °C. Live/dead® Fixable Blue amine-reactive dye (Life Technologies) was used to assess cell viability and exclude dead cells from the analysis. To identify T lymphocytes, macrophages, and granulocytes, cells were stained with monoclonal antibodies for 30 min at 4 °C, washed in PBS/10% FCS, and fixed in CellFIX$^{TM}$ (BD Biosciences). The antibodies were anti-CD45 PerCp (clone B058-1283), anti-CD3 V500 (clone SP34-2), anti-CD4 PE-Cy7 (clone L200), anti-CD8 V450 (clone BW138/80; Miltenyi Biotec), anti-CD11b Alexa Fluor 700 (clone ICRF44), and anti-HLA-DR APC-H7 (clone G46-6) (all from BD Biosciences). Acquisition was performed on a BD LSRII and analyzed using FlowJo 9.8.3 (Tree Star, Ashland, OR). According to

the WHO, leukocytospermia was defined as a leukocyte concentration in the ejaculate $\geq 1 \times 10^6/\text{ml}$[56]. Semen samples were considered to be leukocytospermic if a minimum of 50,000 CD45[+] events/ml were acquired by flow cytometry (consisting of 10,000 events per ejaculate)[30]. This cut-off value was defined as the minimal number of acquired CD45[+] events required to perform statistics on the various leukocyte populations.

For immunocytochemical analysis by microscopy, semen was centrifuged as described above and 150 μl of a 1/100 dilution of the pellet was deposited onto a microscope slide using a cytospin (7 min at $500 \times g$). Staining was performed using the RAL 555 rapid May-Grunwald Giemsa staining kit (Ral Diagnostics, France) and images acquired with a Nikon Eclipse 80i microscope.

**Cytokine quantification in semen.** Inflammatory cytokines/chemokines and TGFβ isoforms were measured in 25 μL seminal plasma using the Milliplex® Map Non-Human Primate Cytokine Magnetic Bead Panel - Premixed 23-plex and TGFβ 1,2,3 Magnetic Bead Kit (Merck Millipore). Immunoassays were performed according to the manufacturer's instructions. Data were acquired using a Bio-Plex 200 instrument and analyzed using Bio-Plex Manager Software, version 6.1 (Bio-Rad).

**Cells, virus, and reagents.** Immature DCs were obtained from CD14[+] cells isolated from human PBMCs and cultured for 5 to 6 days in RPMI 1640 with 1% FCS, 50 ng/ml granulocyte–monocyte colony-stimulating factor (GM-CSF), and 20 ng/ml Interleukin-4 (Peprotech Inc.). DCs were characterized as CD1a[+], CD11c[+], CD4[+], CCR5[+], DC-SING[+], CD14[−], CXCR4[−] cells by flow cytometry. Caco-2 cells (clone E, kindly provided by Dr. Maria Rescigno, Humanitas Institute, Italy) were grown in Dulbecco's modified Eagle's medium (DMEM) with 10% FCS, 1% non-essential amino acids (NEAA), and penicillin (100 U/ml)/streptomycin (100 mg/ml). All reagents were purchased from the Lonza Group (Switzerland) unless otherwise indicated.

SIVmac251 virus used for infection of colorectal explants was kindly provided by Dr. M. Muller Trutwin, Institute Pasteur, and the viral stock was produced in CEMx174 cells grown in RPMI medium. Cells $(30 \times 10^6)$ were incubated at a $10^{-2}$ multiplicity of infection (MOI) for 2 h at 37 °C and the volume adjusted to $1 \times 10^6$ cells/ml. Cultures were maintained for 21 days, changing half of the cells every 3 to 4 days. Viral replication was monitored by measuring p27 levels in culture supernatants by ELISA (RETRO-TEK SIV p27 Ag ELISA kit, Helvetica Health Care). Cell-free viral stock was passed through a 0.2-μm pore-size filter, aliquoted, and stored at −80 °C.

**Polarized explant treatment.** Macaque sigmoid colon was harvested at necropsy, placed in cold PBS containing 100 U penicillin/ml, 100 mg streptomycin/ml, and 50 mg/ml gentamicin and processed within 30 min. After abundant washes in the same solution, specimens were transferred into RPMI supplemented with 10% FCS, 100 U/ml penicillin/streptomycin, 1% glutamine, 1% NEAA, 1% Na-pyruvate, 1% 1 M HEPES buffer (complete medium), and the epithelial surface exposed. A polarized ex vivo culture model was established, slightly modifying our previously published protocol[8]. Briefly, small tissue explants (4.0 mm in diameter), including the epithelium and submucosa, were cut with a biopsy punch (Stiefel, Laboratories, Inc.) and placed on a 24-well hanging insert (3 μm PET, Millicell cell culture insert, Millipore), with the submucosa facing the filter. The explant was surrounded with 3% agarose gel prepared in RPMI and a polystyrene cylinder (I.D. × H 4.7 mm × 8 mm, Sigma-Aldrich) was immediately inserted in the liquid agarose gel to seal the tissue and avoid leakage. The basal chamber was filled with 1 ml complete medium. Mucosal explant cultures were treated apically with viral culture supernatants (100 μl of SIVmac251 stock) or complete medium (negative control) with or without seminal plasma at 1:1 ratio for 2 h (unless otherwise specified). At the end of stimulation, the cylinder was removed, the tissue gently collected, while avoiding damage to the mucosa, and abundantly washed in RPMI. Explants were further subjected to either (1) ex vivo culture, (2) a viability and tightness check, or (3) immunohistochemistry (IHC)/immunofluorescence (IF) as further described. A schematic representation of the experimental protocol is shown in Supplementary Fig. 3.

*(1) Ex vivo tissue culture.* Explants were cultured in pairs, with the epithelium uppermost, supported on a presoaked support (Gelfoam(R) rafts, Pfizer) at the air–media interface in 24-well plates containing 500 μl media. Experiments were performed two or three times in triplicate. Culture supernatant was replaced with fresh complete medium every three days for up to 12 days of culture, collected, and frozen at −80 °C until use. Supernatants were used to measure viral production by RT-qPCR and quantify inflammatory cytokines by Luminex. The basolateral medium was also sampled 24 h after treatment for specific experiments and the cells that migrated from tissues were collected and co-cultured with uninfected CEMx174, as described in Supplementary Information. Explants were collected at the end of the culture, rinsed in PBS, and frozen at −80 °C.

*(2) Tissue explant viability by 1-(4,5-dimethylthiazol-2-yl)-3,5-diphenylformazan (MTT) assay and fluorescein isothiocyanate-dextran leakage assay.* Explants were exposed in triplicate for 2 or 4 h to complete medium (negative control) or seminal plasma (25%, 50% in complete medium), washed, weighed, and incubated in RPMI containing 3-(4,5-dimethyl-2-thiazolyl)-2,5-diphenyltetrazolium bromide (MTT)

substrate (0.5 mg/mL, Sigma-Aldrich) for 3 h at 37 °C. After removal of the MTT solution, tissues were incubated in methanol overnight at room temperature in the dark to dissolve non-specific precipitates. Absorbance of the MTT-formazan product was measured at 570 nm (corrected at 630 nm) using a Spark plate reader (TECAN SPARK 10 M). Tissue viability was determined by dividing the absorbance by the dry weight of the explant. The effect of seminal plasma on tissue viability was calculated as the ratio between seminal plasma-treated explants and negative controls.

A suspension of fluorescein isothiocyanate-dextran (250 μg/ml FD4, 4 kDa; Sigma-Aldrich) was added to the apical surface of the sealed polarized explant and a tight Caco-2 monolayer treated, or not, with seminal plasma (25%, 50% in complete medium). After 2 or 4 h of incubation at 37 °C, the basolateral medium was sampled and FD4 quantified at 520 nm using a Spark plate reader (TECAN SPARK 10 M). The optical density was expressed as the percentage of the positive control (FD4 added to the basal medium at the beginning of the experiment).

*(3) Histopathology analysis, immunofluorescence markers, and confocal microscopy.* Tissues were fixed in 4% PFA for 4 h, cryoprotected in 10% sucrose overnight, embedded in optimal cutting temperature embedding medium (OCT, VWR) in plastic cryomolds (Corning), and snap-frozen. One explant was fixed immediately at baseline to serve as the baseline control. Tissue sections were stained with Hematoxylin/Eosin and the morphology evaluated under a light microscope (Nikon Eclipse 80i). For confocal microscopy analysis of colonic tissues, at least 5 to 6 serially cut 10-μm thick sections, sampled 100 μm apart, were obtained. A list of the antibodies used for confocal microscopy analysis of colonic tissues is provided in Supplementary Table 1. Primary antibodies were used at a final concentration of 10 μg/ml. Tissues were stained with secondary antibody alone (anti-mouse IgG Alexa Flour conjugates, Invitrogen) as a negative control. Single-channel images from z-series were collected from at least three representative fields with a Leica TCS SP8 confocal microscope (Leica Microsystems GmbH, Wetzlar Germany) and the images processed using ImageJ software, as described above.

**Quantification of viral load in explant culture supernatants and real-time PCR for SIV DNA quantification.** The SIV copy number in culture supernatants was determined in 100-fold diluted culture supernatants by quantitative real-time RT-PCR[57], using SIV *gag* primers F 5′-GCAGAGGAGG AAATTACCCAGTAC-3′, R 5′-CAATTTTACCCAGGCATTTAATGTT-3′, probe FAM-5′-TGTCCACCTGC-CATTAAGCCCGA-3′-BHQ1, Superscript III platinum One-Step qRT-PCR system (Invitrogen), and CFX96 thermocycler (Bio-Rad). Reverse transcription was done at +56 °C for 30 min and followed by 5 min of denaturation at +95 °C and by 50 cycles of 15 s at 95 °C and 30 s at 60.3 °C. Calibrated SIVmac251 was used to generate a standard curve and the SIVmac251 *gag* cDNA sequence, ligated into the pCR4-TOPO (Invitrogen) plasmid and purified with the HiSpeed Maxiprep kit (Invitrogen), was used as a positive control. For each qRT-PCR run, standard curve, positive and negative controls, and samples were run in duplicate. Copy numbers were calculated by interpolating CT of samples in the standard curve. The limit of detection was 1000 copies/mL in this setting where 100-fold diluted samples were used. For determination of SIV DNA copy numbers[58], total DNA was extracted from explants frozen after 12 days of culture. Quantitative real-time PCR was performed in duplicate on 500 ng of DNA using SIV *gag* primers and probe as described above and on 50 ng of DNA for *GAPDH* gene using primers F 5′-ATGACCCCTTCATTGGCCTC-3′, R 5′-TCCACGACATACTCAGTGCC-3′, probe FAM-5′-CGAGCTTCCCGTTCTCAGCC-3′-BHQ1. The *GAPDH* gene was used to normalize results per million cells using a standard curve of DNA considering that 1 μg of DNA corresponds to 131,300 cells. SIV *gag* standard curve was generated by dilution of pCR4-TOPO-SIVmac251 *gag* cDNA in DNA of lymph nodes of SIV-negative macaques. SIV DNA copy numbers were calculated by interpolating CT of samples in *gag* standard curve and were normalized using *GAPDH* gene data to be expressed per million cells. The limit of detection is 10 copies per million cells.

**Time-lapse confocal microscopy analysis.** Monoclonal antibody anti-CD45 (clone DO58-123) and its isotype control (IgG1-MOPC-03, both from BD Pharmingen) were covalently conjugated to fluorochromes with AlexaFluor-647, using a microscale protein labeling kit (Life Technology), and the protein and fluorochrome concentrations of the conjugated antibodies measured using a Nanodrop microvolume spectrometer system (Thermo Scientific, Waltham, USA). NucBlue Live Cell Stain, ReadyProbes (Hoechts33342) (Invitrogen) was used to label cell nuclei. A schematic representation of explant treatment is shown in Supplementary Fig. 6. Explants were incubated for 1 h, either with 25 μg/ml of the conjugated CD45 antibody or isotype control, and one drop of the NucBlue nuclear dye in a final volume of 200 μl and then washed in 1X PBS. They were then incubated for 1 h with either complete medium, SIVmac251, or 25% LSP in complete medium in a polarized manner. After extensive washing, the explants were transferred to a six-well plate with complete medium and submitted for video-microscopy. Three different regions of each explant were randomly acquired with a Plan Fluor 20x DIC objective (NA: 0.45) on a Nikon A1R confocal fast-laser scanning system (Nikon Corporation, Japan) equipped with a thermostatic chamber (37 °C, 5% $CO_2$). Images were recorded with a high-speed resonant scanner at 29 min intervals for up to 14 h. Z-series of 54 μm were acquired every 3 μm.

NIS-Elements AR Analysis 5.02.0 (Nikon) and ImageJ software were used to analyze and quantify CD45$^+$ cells. An automatic threshold algorithm was applied to filter the background signal using ImageJ software. The number of fluorescent objects, with a size >4.5 μm, was quantified after binary transformation of each selected frame. Results were represented and analyzed by GraphPad Prism v8.0.2 software.

**Purification of lamina propria mononuclear cells, exposure to seminal plasma and immunophenotyping**. To isolate lamina propria mononuclear cells (LPMCs), colonic sigmoid tissue from uninfected animals was cut into small pieces and incubated for 1 h at 37 °C in HBSS (Fisher Scientific) containing collagenase IV (0.3 mg/ml, Sigma-Aldrich), FCS (5%, Fisher Scientific), and DNase I (5 U/ml, Sigma-Aldrich). Thawed cells from six different donors (1 million/condition) were incubated during 2 h with complete medium (control), 25% NSP pool or 25% LSP pool, then extensively washed and cultured in 96-well plate. Seven hours and 72 h later, DC and lymphocyte phenotyping was performed, respectively. Cells were incubated with the antibodies listed in Supplementary Table 2. Acquisition was performed on a BD Fortessa and analyzed using FlowJo 9.8.3 (Tristar, USA) software. At least 500 events were recorded for rare cell populations (i.e., pDC).

**Co-culture of migrating cells with CEMx174**. Cells migrating from colonic explants after 24 h of culture on a Gelfoam$^{(R)}$ raft (see above) were centrifuged and transferred to a U-bottomed 96-well plate. Then, uninfected CEMx174 cells were added (ratio 3:1 vs. migrated cells) in complete medium. A minimum of 50,000 CEMx174 cells/well was added if the number of migrating cells was <10,000. The culture was carried out in 250 μl complete medium/well, at 37 °C in a standard incubator. CEMx174 cells cultured alone served as a negative control. Half of the culture supernatant was replaced with fresh complete medium every 3 days for up to 12 days of culture. Supernatants were frozen until further analysis. SIV-RNA was quantified (see above).

**Caco-2/DC transwell system**. Caco-2 cells ($1 \times 10^5$) cultured in DMEM plus 10% FCS, were seeded on the upper face of a 6.5-mm filter (3-mm pore Transwell filter, Costar) in a 24-well plate for 7 to 8 days, until a transepithelial resistance (TER) ≥ 330 V/cm$^2$ was achieved. DCs ($4 \times 10^5$) were seeded on the opposite side of the filter for 4 h. Cell-free SIVmac251, a pool of 25% NSP or LSP, or a mixture of SIVmac251 and seminal plasma (ratio 1:1) were added to the apical side of the Caco-2 cells for 1.5 h. Negative controls consisted of medium (DMEM 10% FCS) or supernatant from a uninfected CEMx174 culture. Caco-2/DC filters were fixed in 2% paraformaldehyde (PFA) for successive evaluation by confocal microscopy. A list of the antibodies used for the confocal microscopy analysis of Caco-2 cells and DCs is provided in Supplementary Table 1.

**DC morphometry**. The entity of migrating DCs in the Caco-2/DC co-culture system was evaluated in three representative experiments using specimens treated with either medium only, SIVmac251 (either alone or mixed at a ratio 1:1 with a pool of LSP or NSP), or a NSP and LSP pool (25% dilution). Z-series of images were collected in a single channel with a Leica TCS SP8 confocal microscope from at least three representative fields/specimen. For representation, the images were merged and the 2D free projection max obtained by LAS software (Leica Micro-systems). For DC quantification, images obtained at the center of the Caco-2-cell monolayer were selected from each z-series. Three or four z-series from three different experiments were analyzed. The z-series of images were processed using ImageJ 1.49 v software (National Institute of Mental Health, Bethesda, USA, http://rsb.info.nih.gov/ij/) and the area occupied by DCs expressed as percentage of the total area of the z-series. Results were represented and analyzed by GraphPad Prism v8.0.2 software.

**Statistics and reproducibility**. All data visualization and statistical analyses were carried out using GraphPad Prism v8.0.2 software (GraphPad software, La Jolla, USA) and R v3.5.2. All experiments were performed at least three times, except where otherwise noted.

The non-parametric Spearman rank correlation test was used to investigate the relationship between concentration of cyto/chemokines present in seminal plasma and explants viral replication. Statistical significance between two groups was tested using Wilcoxon rank-sum tests or Wilcoxon signed-rank tests. When comparing more than two groups, Kruskal–Wallis or Friedman tests were used to evaluate the statistical significance and $p$-values were corrected for multiple comparisons using the Benjamini, Krieger, and Yekutieli FDR approach. $P$ values ≤ 0.05 two-tailed tests were considered significant, $*p < 0.05$, $**p < 0.01$, $***p < 0.001$, $****p < 0.0001$.

**Reporting summary**. Further information on research design is available in the Nature Research Reporting Summary linked to this article.

## Data availability

The authors declare that all the data supporting the findings of this study are available within the paper and its supplementary information files. Source data underlying figures is presented in Supplementary Data 1. All other data are available from the corresponding author upon reasonable request.

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

## Acknowledgements

We thank all members of the FlowCyTech, Animal Science and Welfare, and L2I and L3I core facilities of the IDMIT infrastructure for their excellent expertise and outstanding contribution. The SIVmac251 viral strain was kindly provided by Dr. Micaela Muller Trutwin and Dr. Beatrice Jacqueline, Institute Pasteur. This work was funded by the French Agence Nationale de Recherches sur le Sida et les Hépatites Virales (ANRS, decision n° 14415/15516). M.C. was a beneficiary of a Marie Curie Individual fellowship (grant agreement n° 658277 for the project DCmucoHIV). N.T. was supported by fellowships from the ANRS. This work was also supported by the "Programme Investissements d'Avenir" (PIA) managed by the ANR under reference ANR-11-INBS-0008, funding the Infectious Disease Models and Innovative Therapies (IDMIT, Fontenay-aux-Roses, France) infrastructure, and ANR-10-EQPX-02-01, funding the FlowCyTech facility (IDMIT, Fontenay-aux-Roses, France). The funders had no role in study design, data collection or interpretation, or the decision to submit the work for publication.

## Author contributions

Study conception and design: M.C. and R.L.G. Data acquisition: M.C., N.H., S.T., and C.G. Analysis and interpretation of the data: M.C., S.H., N.H., S.T., C.G., N.T., H.H., C.C., and N.D.B. Draft of the manuscript: M.C. Critical revisions: M.C., S.H., N.T., and R.L.G. All authors have read and approved the final version of the manuscript.

## Competing interests

The authors declare no competing interests.
