## [Peer Review File · Communications Biology]

Reviewers' comments:

Reviewer #1 (Remarks to the Author):

The manuscript by Cavarelli et al. entitled "Leukocytospermia induces intraepithelial recruitment of dendritic cells and increases SIV replication in colorectal tissue explants" evaluates the effect of inflammation on SIV transmission from semen to the colorectal tract. This is an important subject that has not been extensively studied before. The authors present very elegant work showing how the increase level of pro-inflammatory cytokines in semen results in increased SIV infection levels in the colorectal tract of macaques.

A few points need to be addressed by the authors:

- The authors show immunocytochemistry images in Figure 1 A and B which are clearly different, but it would help the reader to add a short description of those differences.
- Did the authors observe a difference in pH between LSP and NSP?
- Throughout the paper the focus on LSP is sometimes lost. For example, in lines 108 and 109, the text refers to Figure 3A where results for LSP are shown, however in the text the authors talk about SP. The same comment applies to the last paragraph in page 7, where the authors show how SP increases viral replication, but it would have been more interesting to compare the impact of NSP vs. LSP vs. virus in the absence of semen. I understand that some experiments can certainly not be repeated, and if this is the case, the authors should clearly state this as a limitation of the study in the discussion.
- Could the authors discuss why they see in Figure 4 A and D, significantly higher levels of viral production at day 12 for LSP compared to NSP, but no significant difference is observed at the level of DNA copies/cell? It is also surprising that a significant difference is not detected in the number of infected cells following exposure to NSP vs. LSP taking into account the difference of DC migration shown in Figure 7.
- It would be interesting to further support the conclusion reached in lines 140-142 with some correlation analysis or PCA between inflammatory markers and infectivity levels.
- The section entitled "Characterization of colonic leukocyte populations" does not fit very well there, it might help the reader to move it to the beginning of the Results or otherwise, there should be a sentence helping the reader to understand why this data is shown.
- In line 279 of the Discussion, I would emphasize that the cytotoxicity of SP is laboratory artifact and not a phenomenon observed in nature.
- There is a typo at the end of line 279, where it should read "limitations" instead of "limit".
- In Line 285, all references cited are with human tissue and not just rectal, but colorectal; hence, the authors should correct this sentence to clarify these two points.
- In line 304, the authors mention HIV and I think it would be important to add a reference about the lack of complete repopulation in humans even under treatment.
- Could the authors confirm the % of FCS they used to grow Caco-2 cells, for which 20% and not 10% is usually required.
- Line 445: it should be 24-well hanging insert and not 12- based on Supplementary Figure 3.
- Line 560: the authors should specify which type of DNase they used.

- The number of animals/specimens and replicates for each condition are not very clear or not mentioned in the figure legends of Figures 4 (A and C) and 8.
- In my opinion it would be easier for the reader to compare data between colon and blood shown in Figure 5 A in one merged graph.

Reviewer #2 (Remarks to the Author):

Brief summary of the manuscript

The manuscript by Cavarelli et al. addresses an important topic in the biology of HIV-1 transmission being unprotected anal intercourse one of the major routes of virus acquisition. To date there is limited data on the role of semen and its components in HIV transmission to the intestinal mucosa. Building on available evidence that leukocytospermia is associated with local inflammation in the male genital tract, as reflected by a general increase in pro-inflammatory cytokines in semen, the authors aim to study whether these changes affect virus transmission to the intestinal mucosa. The authors used an experimental system of polarized sigmoid colon tissue explants from non-human primates to model exposure to seminal plasma as well as HIV transmission. The major findings of the manuscript are that exposure to leucocytospermic seminal plasma (LSP) elicits an inflammatory response, as evaluated by tissue transcriptomic and leukocyte migration analysis, and enhances SIV replication locally in the intestinal mucosal, also in comparison with normal seminal plasma (NSP). The authors speculate about the mechanistic link between the observed inflammatory response and enhancement of virus replication without experimentally addressing the role of the identified potential determinants of transmission.

Overall impression of the work

The immunoregulatory effect of semen has been studied mainly in the context of HIV transmission to the female genital mucosa (FGM). In the manuscript by Cavarelli et al., the finding that SP, and in particular LSP, enhances SIV replication in non-human primate colonic explants is novel and relevant. The relevance of the study relies on the polarized nature of tissue explant model used to address the effect of SP on virus replication. Overall the experimental system is well characterized in respect to tissue viability and integrity upon SP treatment, as semen and its components have been reported to be toxic at high concentrations in isolated cell cultures. The setup and results of the SIV infection experiments are convincing although a clarification on the sample size is needed. The tissue transcriptomic analysis offers a comprehensive picture of the changes associated with SP treatment although it could be better integrated in the text. Likewise, although valuable, the phenotypic analysis of tissue leukocytes and the leukocyte migration assays appear as isolated pieces of evidence that offer little support to the manuscript main claim, partly due to the lack of some experimental controls and references to *in vivo* studies, which as mentioned before are rare.

Specific comments, with recommendations for addressing each comment

In general all experiments are clearly described and well executed. I have some remarks on three points:

1. The number of independent experiments of explant infection with SIV in the presence of a pool of SP is 2 (figure 4), but the results are supported by an additional 3 experiments using SP from individual macaques, as reported in figure S4C. I assume that infection experiments were carried out on tissues from a total of at least 3 different animals. Otherwise, I may have some concerns on the impact of inter-individual variability on the results with lower numbers. Please clarify if in figure S4C the infection was carried out using NSP/LSP from 1 animal to infect tissue from 3 different donors. 

2. It is not clear how SP-mediated leukocyte recruitment as shown in figure 7 would contribute to the observed enhancement of SIV replication in colonic explants since the only comparison made in the experiments reported in figure 7 was between medium and LSP treatment. Additional experimental conditions including exposure to the virus inoculum, with and without LSP, should be included since the virus itself may have a chemotactic effect on immune cells as stated in the text. In fact, all the data on the chemotactic properties of SP compared to the virus inoculum, alone or in combination with SP, were produced using Caco-2 cells as shown in figure 6. This co-culture system is a useful complement but the main focus should be on the same system used for SIV infection experiments.

3. In addition to their potential migration associated with SP treatment, it would be informative to replicate the phenotypic analysis carried out on tissue leukocytes, and in particular the target cells of the virus, upon explant treatment with LSP (figure 5). This analysis may offer a mechanistic insight into the observed enhancement of SIV replication by LSP and smoothen the transition from one result section to the other. For instance, increased levels of CCR5 ligands in LSP may be responsible for the recruitment of target cells to the upper layers of the explants downregulating CCR5 at the same time.

General remarks

The authors could further validate the relevance of their experimental model by comparing the transcriptome analysis of SP-treated explants with the results of, to my knowledge, the only in vivo study of the immunological changes associated with unprotected anal intercourse (Kelley, C., et al. The rectal mucosa and condomless receptive anal intercourse in HIV-negative MSM: implications for HIV transmission and prevention. *Mucosal Immunol* 10, 996–1007 (2017). <https://doi.org/10.1038/mi.2016.97>). Despite obvious differences between the two settings, it would be interesting to know whether a Th17 signature, as identified in the work by Kelly et al., is present in SP-treated explants being Th17 cells among the preferential targets of the virus in the female genital mucosa and selectively depleted in the gut in vivo early during infection.

Among the limitations of the study, I would clearly state in the discussion that despite LSP may serve as useful surrogate of the immunological changes associated with local inflammation/infection of the male genital tract, the analysis of LSP carried out by the authors was restricted to pro-inflammatory cytokines. Therefore LSP might not be fully representative of HIV- and possibly SIV-infected SP. The authors may want to comment on the opportunity of using SP from SIV-infected NHPs for a follow-up study.

Minor remarks

Figure 2A x axis. IL- instead of Il-, e.g. IL-15

Line 74. IL-1 β

Line 214 and 238. LSP instead of LPS

Reviewer #3 (Remarks to the Author):

The authors of this manuscript employ cynomolgus macaque colorectal tissue explants to explore the effect of leukocytospermic seminal plasma (LSP) on SIVmac251 infection. The authors argue that LSP enhances SIV infection more efficiently than normal SP (NSP), with the increased viral replication linked to the level of inflammatory and immunomodulatory cytokines found in SP. In very elegant microscopy experiments, the authors show that LSP-induced leukocyte accumulation on the apical side of the colorectal lamina propria and the recruitment of a higher number of CD11c⁺ cells than with NSP. Microarray assays showed differential modulation of genes associated to cell migration and the regulation of cell projection assembly, suggesting that LSP favored the formation of cellular protrusions. In addition, LSP activated genes and pathways that promote inflammation and viral replication. The experimental approach is not entirely novel, with the first study using human colorectal tissue explants and HIV described in 2005, and in 2013 for macaque-derived tissue explants; the authors have published similar studies with HIV. The manuscript sometimes reads as a compilation of several different experiments with no clear rationale, such as description of macaque semen cell or cytokine content, experiments with macaque colorectal tissue explants and virus, experiments with a human cell line, monocyte-derived dendritic cells, and macaque seminal plasma. Additionally, while it is demonstrated that LSP increases replication of SIV in colorectal samples, no attempt is presented as to identify which components may be the ones associated with this increased replication; although the number of potential culprits may be too large to test. Similarly, whether LSP increases virus replication by inducing activation of target cells present in the mucosa is not demonstrated by flow cytometry, but suggested by gene transcription changes identified with analysis tools with very lenient limits of fold change and statistical significance.

Additional comments include:

- When leukocytospermic semen samples are defined as having more than certain number of leukocytes per ml cutoff, it is expected to find that such samples have more cells than semen samples identified as normal. Thus, the justification for having such large Figure 1 is weak; it could be replaced by a supplemental table. Of note, Figure 1C lacks identification of the axes.
- Figure 2 could be shortened, since it shows the same data as Supplemental Figure 2, represented in different ways. The significance of the colorful Z-score (Figure 2C) is not explained.
- Figure 4A-C show the same data, represented in 3 different forms. Figure 4B is a column graph, not a histogram. One viral RNA figure should be sufficient along with the SIV DNA load.
- Figure 5A compares leukocytes content in the peripheral blood and colon of several animals. What is the relevance of these data to the purpose of the study? Similarly, Figure B-D are descriptive of cells before any exposure to LSP or NSP.
- What is the meaning of each dot for Figure 6B? Was there a control of tissue exposed to NSP?
- Figure 8, gene expression analysis. The large number of differentially expressed genes identified by this analysis may be due to the selection of a small fold-change of 1.2 and lack of False Discovery Rate (FDR)

statistical analysis. What is the biological significance of the large number of downregulated genes for normal seminal plasma compared to medium only?

- Figure S3: There is no A, B, or C designation.

- Figure S4: the virus dose tested varied just by 2-fold. Were there other virus ranges tested? For figure S4B the viral titers seem to be on the very low end, compared with figures S4A and S4C.

- Co-culture assays with CEMx174 cells must have been done in 250 μ l, not 250 ml as presented.

Response to referees'

Reviewers' comments:

Reviewer #1 (Remarks to the Author):

The manuscript by Cavarelli et al. entitled “Leukocytospermia induces intraepithelial recruitment of dendritic cells and increases SIV replication in colorectal tissue explants” evaluates the effect of inflammation on SIV transmission from semen to the colorectal tract. This is an important subject that has not been extensively studied before. The authors present very elegant work showing how the increase level of pro-inflammatory cytokines in semen results in increased SIV infection levels in the colorectal tract of macaques.

A few points need to be addressed by the authors:

- The authors show immunocytochemistry images in Figure 1 A and B which are clearly different, but it would help the reader to add a short description of those differences.

Reply: We have added a description of the differences between panel A and B in the Figure legend. It reads: “Representative immunocytochemical staining of cells from (A) normal semen (NS) and (B) leukocytospermic semen (LS). Numerous spermatozoa (black arrowhead) were present in NS. LS was rich in white blood cells, including lymphocytes (black *), macrophages (red arrowhead) and neutrophils (red *), frequently forming aggregates (I) with spermatozoa.”

- Did the authors observe a difference in pH between LSP and NSP?

Reply: No difference was observed, the pH ranged between 7.2 and 7.4. We added this information in the “methods” section, line 423.

- Throughout the paper the focus on LSP is sometimes lost. For example, in lines 108 and 109, the text refers to Figure 3A where results for LSP are shown, however in the text the authors talk about SP. The same comment applies to the last paragraph in page 7, where the authors show how SP increases viral replication, but it would have been more interesting to compare the impact of NSP vs. LSP vs. virus in the absence of semen. I understand that some experiments can certainly not be repeated, and if this is the case, the authors should clearly state this as a limitation of the study in the discussion.

Reply: we have refocused the attention on LSP when only LSP was used (ex. lines 108-109 and elsewhere and last paragraph of page 7 about time-lapse confocal microscopy). We agree with the reviewer that it would be interesting to compare LSP vs NSP vs SIV in time-lapse confocal microscopy experiments, however we were restricted by the size of the sigmoid colon and forced to make choices about the condition to test. LSP vs SIV comparison is shown in the revised version, whereas NSP was not assayed in this type of experiment. Nevertheless, LSP and NSP (either alone or in combination with SIV) were used on explants when assessing DC migration. In the revised manuscript we have clarified these aspects and discussed the limitations (“Results” lines 176-177, 187-189, and “Discussion” lines 321-324, 329-333).

- Could the authors discuss why they see in Figure 4 A and D, significantly higher levels of viral production at day 12 for LSP compared to NSP, but no significant difference is observed at the level of DNA copies/cell? It is also surprising that a significant difference is not detected in the number of infected cells following exposure to NSP vs. LSP taking into account the difference of DC migration shown in Figure 7.

Reply: Viral load in explant supernatant is the result of cumulative viral production during the 12 days of follow-up. The level of DNA copies/million cells was measured at the end of the culture (day 12). We showed in the Supplemental material that a number of cells (presumably APCs) exit the explant as soon as 24h of explant culture. We do not exclude the possibility that a higher number of SIV infected cells might have exited explants upon treatment with LSP compared to NSP, thus explaining the lack of significant difference in proviral load. This explanation has been added in the Discussion, lines 298-302.

- It would be interesting to further support the conclusion reached in lines 140-142 with some correlation analysis or PCA between inflammatory markers and infectivity levels.

Reply: we have performed a correlation analysis using the non-parametric Spearman rank correlation test to investigate the relationship between concentration of cyto/chemokines present in SP and explants viral replication. The Results are shown in lines 141-146 and Table 1 of the revised manuscript.

- The section entitled “Characterization of colonic leukocyte populations” does not fit very well there, it might help the reader to move it to the beginning of the Results or otherwise, there should be a sentence helping the reader to understand why this data is shown.

Reply: to answer to the comments of the other reviewers and to better integrate these data in the manuscript, this paragraph has been modified and the revised manuscript shows now comparison of T cell subsets population and cells activation before and after exposure to seminal plasma.

- In line 279 of the Discussion, I would emphasize that the cytotoxicity of SP is laboratory artifact and not a phenomenon observed in nature.

Reply: we clarified this aspect in line 284 of the Discussion.

- There is a typo at the end of line 279, where it should read “limitations” instead of “limit”.

Reply: we corrected the typo.

- In Line 285, all references cited are with human tissue and not just rectal, but colorectal; hence, the authors should correct this sentence to clarify these two points.

Reply: we clarified this aspect and added references.

- In line 304, the authors mention HIV and I think it would be important to add a reference about the lack of complete repopulation in humans even under treatment.

Reply: we added the reference of a study performed in untreated and treated HIV patients (line 312 of the revised manuscript).

- Could the authors confirm the % of FCS they used to grow Caco-2 cells, for which 20% and not 10% is usually required.

Reply: in our hands, and as we previously published (ref 8) Caco-2 cells grow well in 10% FCS. We added this information in the Methods, line 607.

- Line 445: it should be 24-well hanging insert and not 12- based on Supplementary Figure 3.

Reply: the reviewer is right, we corrected the mistake.

- Line 560: the authors should specify which type of DNase they used.

Reply: we included this information in the Methods, line 597.

- The number of animals/specimens and replicates for each condition are not very clear or not mentioned in the figure legends of Figures 4 (A and C) and 8.

Reply: Figure legend 4 of the original manuscript reads “Results are shown as the mean \pm SEM of triplicates from two independent experiments”. Since the sentence was reported after the description of panel D, it was possibly not clear that it referred to all panels shown in figure 4. In the revised manuscript we clarify that “Results shown in panels A-C are the mean \pm SEM of triplicates from two independent experiments” (please note that panel D is no longer shown, according to the request of reviewer #3).

Also, Figure legend 8 reads “Explants were exposed to normal seminal plasma... in two independent experiments”. In the revised manuscript we further clarify that each condition was tested in duplicate in each experiment.

- In my opinion it would be easier for the reader to compare data between colon and blood shown in Figure 5 A in one merged graph.

Reply: According to the comments of reviewer #3, blood data are not shown in the revised manuscript.

Reviewer #2 (Remarks to the Author):

Brief summary of the manuscript

The manuscript by Cavarelli et al. addresses an important topic in the biology of HIV-1 transmission being unprotected anal intercourse one of the major routes of virus acquisition. To date there is limited data on the role of semen and its components in HIV transmission to the intestinal mucosa. Building on available evidence that leukocytospermia is associated with local inflammation in the male genital tract, as reflected by a general increase in pro-inflammatory cytokines in semen, the authors aim to study whether these changes affect virus transmission to the intestinal mucosa. The authors used an experimental system of polarized sigmoid colon tissue explants from non-human primates to model exposure to seminal plasma as well as HIV transmission. The major findings of the manuscript are that exposure to leucocytospermic seminal plasma (LSP) elicits an inflammatory response, as evaluated by tissue transcriptomic and leukocyte migration analysis, and enhances SIV replication locally in the intestinal mucosal, also in comparison with normal seminal plasma (NSP). The authors speculate about the mechanistic link between the observed inflammatory response and enhancement of virus replication without experimentally addressing the role of the identified potential determinants of transmission.

Overall impression of the work

The immunoregulatory effect of semen has been studied mainly in the context of HIV transmission to the female genital mucosa (FGM). In the manuscript by Cavarelli et al., the finding that SP, and in particular LSP, enhances SIV replication in non-human primate colonic explants is novel and relevant. The relevance of the study relies on the polarized nature of tissue explant model used to address the effect of SP on virus replication. Overall the experimental system is well characterized in respect to tissue viability and integrity upon SP treatment, as semen and its components have been reported to be toxic at high concentrations in isolated cell cultures. The setup and results of the SIV infection experiments are convincing although a clarification on the sample size is needed. The tissue transcriptomic analysis offers a comprehensive picture of the changes associated with SP treatment although it could be better integrated in the text. Likewise, although valuable, the phenotypic analysis

of tissue leukocytes and the leukocyte migration assays appear as isolated pieces of evidence that offer little support to the manuscript main claim, partly due to the lack of some experimental controls and references to *in vivo* studies, which as mentioned before are rare.

Specific comments, with recommendations for addressing each comment

In general all experiments are clearly described and well executed. I have some remarks on three points:

1. The number of independent experiments of explant infection with SIV in the presence of a pool of SP is 2 (figure 4), but the results are supported by an additional 3 experiments using SP from individual macaques, as reported in figure S4C. I assume that infection experiments were carried out on tissues from a total of at least 3 different animals. Otherwise, I may have some concerns on the impact of inter-individual variability on the results with lower numbers. Please clarify if in figure S4C the infection was carried out using NSP/LSP from 1 animal to infect tissue from 3 different donors.

Reply: The reviewer is right, experiments with the pool of SP were carried out on explants from two different donors and the experiments with the individual SP on explants from three additional donors, for a total of five different independent donors.

We have clarified this point in the Results (lines 127-129) and in the legend of Supplementary figure 4 of the revised manuscript.

2. It is not clear how SP-mediated leukocyte recruitment as shown in figure 7 would contribute to the observed enhancement of SIV replication in colonic explants since the only comparison made in the experiments reported in figure 7 was between medium and LSP treatment. Additional experimental conditions including exposure to the virus inoculum, with and without LSP, should be included since the virus itself may have a chemotactic effect on immune cells as stated in the text. In fact, all the data on the chemotactic properties of SP compared to the virus inoculum, alone or in combination with SP, were produced using Caco-2 cells as shown in figure 6. This co-culture system is a useful complement but the main focus should be on the same system used for SIV infection experiments.

Reply: We believe that the reviewer inverted Figure 7 with Figure 6 in his/her comment.

In Figure 7A we already showed comparison of medium with virus, LSP and SIV+LPS using explants, thus the same system used for SIV infection experiments, and not only Caco-2 cells as the reviewer state. Moreover, to give a more comprehensive picture of the different chemotactic abilities of the virus, NSP and LSP, in the revised manuscript we show also results of explants exposed to NSP and SIV+NSP. In summary, all the possible combination of explant stimulation (medium, SIV, NSP, LSP, SIV+NSP, SIV+NSP) is shown.

Figure 6 instead, showed time lapse confocal microscopy experiment comparing medium vs LSP and measuring total CD45+ cells migration. In the revised manuscript we show also results with SIV, as the reviewer asked, and we confirmed that the virus has a chemotactic effect on immune cells in this experimental model. While in this experimental setting the condition SIV+LSP was not used (we were restricted by the size of the sigmoid colon and forced to make choices), we believe that the experiments presented in Figure 7 (which include also the condition SIV + LSP) should answer the reviewer's concerns.

In the revised manuscript we have clarified these aspects and discussed the limitations ("Results" lines 176-177, 187-189, and "Discussion" lines 321-324, 329-333).

3. In addition to their potential migration associated with SP treatment, it would be informative to replicate the phenotypic analysis carried out on tissue leukocytes, and in particular the target cells of the virus, upon explant treatment with LSP (figure 5). This analysis may offer a mechanistic insight into the observed enhancement of SIV replication by

LSP and smoothen the transition from one result section to the other. For instance, increased levels of CCR5 ligands in LSP may be responsible for the recruitment of target cells to the upper layers of the explants downregulating CCR5 at the same time.

Reply: To address the reviewer's concern, in the revised manuscript we compare the phenotype of the target cells of the virus before and after exposure to NSP and LSP. We exposed lamina propria mononuclear cells (LPMC) from the sigmoid colon to medium, NSP and LSP, and performed flow cytometry analysis of lymphocytes (results are shown in Figure 5) and DCs. As hypothesized by the reviewer, we observed significantly lower levels of CCR5+ DCs exposed to LSP when compared to NSP, possibly due to the increased levels of CCR5 ligands in LSP. These results are shown in the Results lines 203-213 and Figure 7E, and Discussion lines 339-342.

General remarks

1) The authors could further validate the relevance of their experimental model by comparing the transcriptome analysis of SP-treated explants with the results of, to my knowledge, the only *i>* study of the immunological changes associated with unprotected anal intercourse (Kelley, C., et al. The rectal mucosa and condomless receptive anal intercourse in HIV-negative MSM: implications for HIV transmission and prevention. *Mucosal Immunol* 10, 996–1007 (2017). <https://doi.org/10.1038/mi.2016.97>). Despite obvious differences between the two settings, it would be interesting to know whether a Th17 signature, as identified in the work by Kelly et al., is present in SP-treated explants being Th17 cells among the preferential targets of the virus in the female genital mucosa and selectively depleted in the gut *i>* early during infection.

Reply: We added a section in the Discussion, lines 366-370, to compare our transcriptome analysis results with those obtained by Kelley et al. It reads: “Our results are also in agreement with finding from a previous study conducted in HIV negative men who have sex with men (MSM) undergoing condomless receptive anal intercourse, which reported differential expression of genes implicated in mucosal injury and repair and immune activation. However, this study reported of a Th17 signature which instead did not come out in our study, possibly because of differences in the experimental approach used”.

2) Among the limitations of the study, I would clearly state in the discussion that despite LSP may serve as useful surrogate of the immunological changes associated with local inflammation/infection of the male genital tract, the analysis of LSP carried out by the authors was restricted to pro-inflammatory cytokines. Therefore, LSP might not be fully representative of HIV- and possibly SIV-infected SP. The authors may want to comment on the opportunity of using SP from SIV-infected NHPs for a follow-up study.

Reply: We agree with the reviewer on the importance of using infected semen in future studies and have discussed this point in the revised manuscript, lines 385-388: “Moreover, while LSP from uninfected macaques is enriched in pro-inflammatory molecules, yet this complex environment might not be fully representative of SIV-infected SP. Future NHP studies or cohort studies of HIV-1 patients evaluating the effect of infected semen are warranted”.

Minor remarks

Figure 2A x axis. IL- instead of Il-, e.g. IL-15

Line 74. IL-1 β

Line 214 and 238. LSP instead of LPS

Reply: these errors have been corrected

Reviewer #3 (Remarks to the Author):

The authors of this manuscript employ cynomolgus macaque colorectal tissue explants to explore the effect of leukocytospermic seminal plasma (LSP) on SIVmac251 infection. The authors argue that LSP enhances SIV infection more efficiently than normal SP (NSP), with the increased viral replication linked to the level of inflammatory and immunomodulatory cytokines found in SP. In very elegant microscopy experiments, the authors show that LSP-induced leukocyte accumulation on the apical side of the colorectal lamina propria and the recruitment of a higher number of CD11c+ cells than with NSP. Microarray assays showed differential modulation of genes associated to cell migration and the regulation of cell projection assembly, suggesting that LSP favored the formation of cellular protrusions. In addition, LSP activated genes and pathways that promote inflammation and viral replication. The experimental approach is not entirely novel, with the first study using human colorectal tissue explants and HIV described in 2005, and in 2013 for macaque-derived tissue explants; the authors have published similar studies with HIV. The manuscript sometimes reads as a compilation of several different experiments with no clear rationale, such as description of macaque semen cell or cytokine content, experiments with macaque colorectal tissue explants and virus, experiments with a human cell line, monocyte-derived dendritic cells, and macaque seminal plasma. Additionally, while it is demonstrated that LSP increases replication of SIV in colorectal samples, no attempt is presented as to identify which components may be the ones associated with this increased replication; although the number of potential culprits may be too large to test. Similarly, whether LSP increases virus replication by inducing activation of target cells present in the mucosa is not demonstrated by flow cytometry, but suggested by gene transcription changes identified with analysis tools with very lenient limits of fold change and statistical significance.

Additional comments include:

- When leukocytospermic semen samples are defined as having more than certain number of leukocytes per ml cutoff, it is expected to find that such samples have more cells than semen samples identified as normal. Thus, the justification for having such large Figure 1 is weak; it could be replaced by a supplemental table. Of note, Figure 1C lacks identification of the axes.

Reply: we believe that the information shown in Figure 1 are useful to the reader and that a table would be less informative. According to the comments of reviewer #1, we further implemented panel A to better explains the difference between normal and leukocytospermic semen. Axes of Figure 1C have been fixed.

- Figure 2 could be shortened, since it shows the same data as Supplemental Figure 2, represented in different ways. The significance of the colorful Z-score (Figure 2C) is not explained.

Reply: Figure 2 has been shortened (panel A has been eliminated and cytokine concentrations are now shown only in Supplementary Figure 1). We also clarified the significance of the colorful Z-score in Figure legend, line 868. The sentence reads "Data are shown as relative molecule concentration compared to the mean value. Upregulated molecules are shown in red and downregulated molecules in blue".

- Figure 4A-C show the same data, represented in 3 different forms. Figure 4B is a column graph, not a histogram. One viral RNA figure should be sufficient along with the SIV DNA load.

Reply: We reduced the number of panels, we eliminated one graph, (panel B in the previous version) but we kept the difference in fold change at day 12 pi (new panel B) because we believe that this information is complementary to the data shown in panel A.

- Figure 5A compares leukocytes content in the peripheral blood and colon of several animals. What is the relevance of these data to the purpose of the study? Similarly, Figure B-D are descriptive of cells before any exposure to LSP or NSP.

Reply: To answer to this comment and to the reviewer's general assessment of the manuscript (*"Similarly, whether LSP increases virus replication by inducing activation of target cells present in the mucosa is not demonstrated by flow cytometry, but suggested by gene transcription changes identified with analysis tools with very lenient limits of fold change and statistical significance"*), and with the aim to better integrate these flow cytometry data in the manuscript, this paragraph has been modified and the revised manuscript shows now comparison of T cell subsets population and cells activation before and after exposure to seminal plasma. Data of blood cells are not shown in the revised manuscript. We exposed lamina propria mononuclear cells (LPMC) from the sigmoid colon to medium, NSP and LSP, and performed flow cytometry analysis of lymphocytes. The results are shown in Figure 5, Results paragraph "Exposure to seminal plasma activates lamina propria lymphocytes", lines 150-170 and Discussion lines 313-320. We observed activation of T lymphocytes exposed to both NSP and LSP after 72h of culture, thus indicating that T cell activation alone do not explain the enhanced replication observed with LSP. We discussed the relevance and limitations of these results: "Of note, our immunophenotype analysis was restricted to 72h post culture. Accordingly, a significative difference in viral replication was not observed at 3 dpi in explant exposed to NSP and LSP, which instead became evident at 9 dpi. It is possible that the complex mucosal environment of the explants responding to NSP vs LSP could differentially affect expression of CD4+ T cell activation markers at this later time points, and further experiments are required to substantiate our observations".

- What is the meaning of each dot for Figure 6B? Was there a control of tissue exposed to NSP?

Reply: we have clarified the meaning of the dot plot in Figure6 legend. The sentence reads: "Each dot represents the number of cells/mm² imaged in the z-stack".

In this specific experiment a control of tissue exposed to NSP was not present. However, NSP control was included on explants analyzed by confocal microscopy to assess DC migration.

- Figure 8, gene expression analysis. The large number of differentially expressed genes identified by this analysis may be due to the selection of a small fold-change of 1.2 and lack of False Discovery Rate (FDR) statistical analysis. What is the biological significance of the large number of downregulated genes for normal seminal plasma compared to medium only?

Reply: We agree with the reviewer that a cut-off of 1.5 would have been a more stringent criteria to look at DEG and we are aware that the down- or up- regulations of gene expressions that we observed are small even if they remain statistically significant (p-value < 0.05). Despite the methodologic limits, we have to point out that the transcriptomics approach we used was exploratory, it was designed to extend the observations that we made using other techniques and it is not the main focus of this work, and certainly the tendency in gene expression we observed need to be confirmed in future studies. This analysis was performed on tissues from a limited number of donors (we were restricted in performing euthanasia of healthy macaques, also because of ethical reasons. Moreover, when tissues were available, we were restricted by the size of the sigmoid color to prioritize the conditions and the

experimental system) and on the bulk population, meaning that we are working with a very heterogeneous population, which can justify a threshold of 1.2. Indeed, the choice of a 1.2 threshold is not unusual in this kind of studies (see for instance Alvarez-Rodriguez et al. Scientific Reports 2020; Marlin et al. Frontiers Immun 2019; Martinez CA et al. Frontiers Veterinary Science 2019).

Out of note, we were not able to identify many differentially expressed genes when applying higher fold-change thresholds. Nevertheless, small variations in terms of fold-increase or fold-decrease can have drastic impacts on associated mechanisms, especially when dealing with regulation mechanisms.

In relation to comment regarding the lack of False Discovery Rate (FDR) approach, we think that there are two main groups of thinkers: those that feel that a correction for multiple tests is needed for transcriptomic analyses and those that think it is not.

When performing multiple statistical tests, incorrect rejections of the (null) hypotheses can occur only by chance. The aim of multiple comparisons corrections is to reduce the number of false positives (also called type I error). Such corrections are important when generating false positive results is a major issue, as for instance concluding that a drug has potential effect while there is actually no effect. A side effect of correcting for multiple comparisons is the increase of the number of false negatives (also called type II error). A type II error can be the incorrect prediction of a molecule to act as a biomarker for a specific pathology.

In this study, and taking into consideration the limited number of biological samples available, we did not apply a False Discovery Rate (FDR) approach and used a fold-change cutoff of 1.2 to detect small significant variations of gene expressions. For the reviewer knowledge, we were not able to identify any differentially expressed genes in our different comparisons when applying a False Discovery Rate (FDR). Therefore, we agree that our transcriptomics approach is exploratory, and made to provide novel hypotheses about the triggered molecular mechanisms in our conditions. We have better explained this point in the revised manuscript (line 218) and discussed the limitation of the work, lines 383-389 of the Discussion.

As pointed by the reviewer a large number of the genes found to be statistically differentially expressed are downregulated for normal seminal plasma compared to medium only. This observation reinforces our believe that these observations are linked to real biological mechanisms. Otherwise, these differentially expressed genes would have been uniformly distributed into down- and up-regulated genes.

Concerning the potential biological significance, we added a section in the Discussion, lines 375-382. The section reads: “Interestingly, mostly downregulated genes were observed in NSP-treated explants, including those linked to immune responses and cytokine production. These results are not surprising since the immune suppressive nature of semen is very well known and is important for creating a tolerogenic environment at the FRT and promoting successful implantation and pregnancy 65. High level of TGF- β found in SP inhibit NK-cell activity by repressing the mTOR pathway 66. In the context of HIV infections, a recent study demonstrated that high concentrations of SP inhibit monocyte phagocytosis and anti-HIV-1 Fc-dependent functions of both neutrophils and monocytes 67”.

- Figure S3: There is no A, B, or C designation.

Reply: we added the designation

- Figure S4: the virus dose tested varied just by 2-fold. Were there other virus ranges tested?

For figure S4B the viral titers seem to be on the very low end, compared with figures S4A and S4C.

Reply: no other virus concentration was tested. Figure S4B shows results of the coculture of cells that exit the tissue with CEMx174, whereas S4A and S4C show direct infection of the tissue. Thus, the viral titers between S4B and S4A/S4C cannot be directly compared.

- Co-culture assays with CEMx174 cells must have been done in 250 ul, not 250 ml as presented.

Reply: we fixed it.

Reviewers' comments:

Reviewer #1 (Remarks to the Author):

The authors have addressed all the points raised by the reviewers.

Reviewer #2 (Remarks to the Author):

In the revised manuscript the authors tried to address the mechanisms behind the observed increase in SIV replication associated with treatment of colon explants with leucocytospermic seminal plasma (LSP) compared with normal seminal plasma (NSP) by adding new data to the analysis of leukocyte migration within explants (figure 6 and 7), and by performing a new set of experiments on lamina propria mononuclear cells (LPMCs) (figure 5). The authors improved the manuscript by clarifying some points, however the manuscript still reads inconsistent in some parts and important data are missing.

Main remarks

1. The data on leukocyte recruitment in SIV-infected explants now showed in figure 6 are a useful addition, although I still feel that it is difficult to draw any conclusion on the combined effect of SP and SIV on leukocyte or DCs migration within explants, as well as the comparison between LSP and NSP, based on the data presented in the revised manuscript. Figure 7A and 7B weakly support the authors' claim as no quantitative analysis was performed to compare the number of migrated DCs between explant treatment conditions.
2. The data in figure 5 are potentially interesting, although I was wondering why the authors decided to carry out SP treatment experiments for cell phenotypic analysis using isolated cells (LPMCs) instead of tissue explants. As the authors state in the discussion, there are important differences between isolated cells and tissue explants in their response to virtually any stimuli. Therefore, it seems contradictory to expect that the data on LPMCs shown in figure 5 may help interpreting the increased SIV replication observed in LSP- vs NSP-treated explants, unless this effect can be recapitulated in LPMC infection experiments.

Minor remarks

1. I was wondering whether the authors measured CCR5 expression levels on colon CD4 T cells, as done for DCs in figure 7E, and the expression of HLA-DR on DCs upon exposure to LSP vs NSP. For consistency, I would consider integrating figure 7E into figure 5.
2. I would consider keeping the results of the phenotypic analysis of colon leukocytes shown in figure 5 of the original manuscript as a supplementary figure, along with representative dot plots illustrating the gating strategy.

Cavarelli et al. Rebuttal letter

Reviewers' comments:

Reviewer #1 (Remarks to the Author):

The authors have addressed all the points raised by the reviewers.

Reviewer #2 (Remarks to the Author):

In the revised manuscript the authors tried to address the mechanisms behind the observed increase in SIV replication associated with treatment of colon explants with leucocytospermic seminal plasma (LSP) compared with normal seminal plasma (NSP) by adding new data to the analysis of leukocyte migration within explants (figure 6 and 7), and by performing a new set of experiments on lamina propria mononuclear cells (LPMCs) (figure 5). The authors improved the manuscript by clarifying some points, however the manuscript still reads inconsistent in some parts and important data are missing.

Main remarks

1. The data on leukocyte recruitment in SIV-infected explants now showed in figure 6 are a useful addition, although I still feel that it is difficult to draw any conclusion on the combined effect of SP and SIV on leukocyte or DCs migration within explants, as well as the comparison between LSP and NSP, based on the data presented in the revised manuscript. Figure 7A and 7B weakly support the authors' claim as no quantitative analysis was performed to compare the number of migrated DCs between explant treatment conditions.
2. The data in figure 5 are potentially interesting, although I was wondering why the authors decided to carry out SP treatment experiments for cell phenotypic analysis using isolated cells (LPMCs) instead of tissue explants. As the authors state in the discussion, there are important differences between isolated cells and tissue explants in their response to virtually any stimuli. Therefore, it seems contradictory to expect that the data on LPMCs shown in figure 5 may help interpreting the increased SIV replication observed in LSP- vs NSP-treated explants, unless this effect can be recapitulated in LPMC infection experiments.

Minor remarks

1. I was wondering whether the authors measured CCR5 expression levels on colon CD4 T cells, as done for DCs in figure 7E, and the expression of HLA-DR on DCs upon exposure to LSP vs NSP. For consistency, I would consider integrating figure 7E into figure 5.
2. I would consider keeping the results of the phenotypic analysis of colon leukocytes shown in figure 5 of the original manuscript as a supplementary figure, along with representative dot plots illustrating the gating strategy.

Reply: We thank the reviewer for the comments and suggestions that strength the manuscript. Please note that the changes in the revised version of the manuscript are highlighted in yellow.

1) We have quantified the number of migrated DCs in explant treatment conditions. The results are shown in Figure 7 C,D and they confirm what we already showed with the Caco-2 cell system, i.e. a statistical significant difference in the number of intraepithelial DC in explants exposed to SIV+LSP (but not NSP) compared to virus alone. The Caco-2/DC experiments seem to us no longer needed in the main manuscript and we suggest to move them in the Supplemental material (new Supplemental Figure 7), along with the related Material and Methods section.

2) We agree with the reviewer that we should be more circumspect in our interpretation of finding obtained using LPMC. Thus, we have updated the discussion and highlighted the need to further confirm with explant systems the results we obtained with LPMC (lines 301-304). Since we do not have left explants, it is not possible to address receptor expression using this type of model.

At variance with previous studies using PBMC as surrogate of mucosal immune cells (such as Introini et al. PlosPathogens 2017), the LPMCs were derived from the colon of the same animals used for explant cultures, thus representing, in our opinion, the most suitable system to obtain reliable results. Indeed, the data on T cell activation we show are consistent with infection results obtained with explants at the same time point of analysis (lines 275-277), thus further substantiating the model.

The assessment of infection by LPMC would certainly be of interest, however it sounds not feasible to us, as it implies to collect intestinal tissues from healthy macaques, and to set-up a large amount of new experimental conditions, including viral titration, incubation time, reagent concentrations etc, using primary intestinal cells.

Minor comments:

1) CCR5 expression by CD4+ T cells was assessed and results shown in the revised version of the manuscript, Figure 7F. No changes were observed following exposure to NSP and LSP. Concerning HLA-DR expression by DC, lamina propria DC have been identified as lineage negative, HLA-DR+ cells. Analysis of MFI revealed no significative changes in HLA-DR expression following incubation with NSP and LSP. This result is not adding relevant information and thus it has not been included in the manuscript.

We understand the reviewer's suggestion to show all the results obtained with LPMC in the same figure. However, we believe it is important to first show DC migration and then CCR5 downregulation as a mechanism supporting this observation.

2) We included a new Supplementary figure 5 showing the gating strategy used to phenotypically characterize T cells and DC. Mostly of the data shown in the figure 5 of the original manuscript are present in the new version, just rearranged to compare the phenotype of cells before and after exposure to seminal plasma. The comparison with blood was removed as asked by the other reviewers.

Sincerely,

Mariangela Cavarelli on behalf of all authors

REVIEWERS' COMMENTS:

Reviewer #2 (Remarks to the Author):

The authors adequately addressed all comments and strengthened their claims by adding the requested analysis. Just one last thing: Line 181. Figure 6B instead of 5B